# Simple Stepsize for Quasi-Newton Methods with Global Convergence Guarantees

## Abstract

Quasi-Newton methods are widely used for solving convex optimization problems due to their ease of implementation, practical efficiency, and strong local convergence guarantees. However, their global convergence is typically established only under specific line search strategies and the assumption of strong convexity. In this work, we extend the theoretical understanding of Quasi-Newton methods by introducing a simple stepsize schedule that guarantees a global convergence rate of $\mathcal{O}(1/k)$ for the convex functions. Furthermore, we show that when the inexactness of the Hessian approximation is controlled within a prescribed relative accuracy, the method attains an accelerated convergence rate of $\mathcal{O}(1/k^2)$ – matching the best-known rates of both Nesterov's accelerated gradient method and cubically regularized Newton methods. We validate our theoretical findings through empirical comparisons, demonstrating clear improvements over standard Quasi-Newton baselines. To further enhance robustness, we develop an adaptive variant that adjusts to the function's curvature while retaining the global convergence guarantees of the non-adaptive algorithm.

## 1 Introduction

Quasi-Newton (QN) methods are among the most widely used algorithms for solving optimization problems in scientific computing and, in particular, machine learning. A prominent example is L-BFGS (Liu and Nocedal, 1989; Nocedal, 1980), a popular Quasi-Newton variant that serves as the default optimizer for logistic regression in the *scikit-learn* library (Pedregosa et al., 2011). These methods implement Newton-like steps, enjoying fast empirical convergence and solid theoretical foundations by maintaining the second-order Hessian approximation $\mathbf{B}_x$ (or its inverse $\mathbf{H}_x = \mathbf{B}_x^{-1}$). For the unconstrained minimization problem of the convex function $f : \mathbb{R}^d \to \mathbb{R}$,

$$\min_{x \in \mathbb{R}^d} f(x), \tag{1}$$

the generic Quasi-Newton update with stepsize $\eta_k > 0$ takes the form

$$x_{k+1} \stackrel{\text{def}}{=} x_k - \eta_k \mathbf{H}_k \nabla f(x_k). \tag{2}$$

The rich history of Quasi-Newton methods can be traced back to methods DFP (Davidon, 1959; Fletcher, 2000), BFGS (Broyden, 1970; Fletcher, 1970; Goldfarb, 1970; Shanno, 1970; Byrd et al., 1987), and SR1 (Conn et al., 1991; Khalfan et al., 1993), which became classics due to their simplicity and practical effectiveness. These approaches build (inverse) Hessian approximations based on *curvature pairs* $(s_k, y_k)$ capturing iterate and gradient differences, $s_k = x_k - x_{k-1}, y_k = \nabla f(x_k) - \nabla f(x_{k-1})$. The stepsize is typically chosen to be unitary $\eta_k = 1$, and this large stepsize is one of the reasons why these classical methods exhibit only local convergence[1]. Global convergence guarantees of Quasi-Newton methods were usually based on the strong convexity assumption and obtained by incorporating linesearches or trust-region frameworks (Powell, 1971; Dixon, 1972; Powell, 1976; Conn et al., 1991; Khalfan et al., 1993; Byrd et al., 1996), yet the obtained convergence guarantees were asymptotic without explicit rates. In particular, for minimizing smooth convex functions, it has been shown that classical Quasi-Newton methods such as BFGS converge asymptotically (Byrd et al., 1987; Powell, 1972).

---

[1]Similarly to the classical Newton method, which can also diverge when initialized far from the solution.

Recent advances in Quasi-Newton methods have primarily focused on addressing these key limitations: the lack of explicit convergence rates for local convergence (Scheinberg and Tang, 2016; Rodomanov and Nesterov, 2021a;b; Lin et al., 2021; Jin et al., 2022; Jin and Mokhtari, 2023; Ye et al., 2023), global convergence (Scheinberg and Tang, 2016; Ghanbari and Scheinberg, 2018; Berahas et al., 2022; Kamzolov et al., 2023; Jin et al., 2024a; Scieur, 2024; Jin et al., 2024b; Wang et al., 2024), and the reliance on the strong convexity assumption (Scheinberg and Tang, 2016; Ghanbari and Scheinberg, 2018; Berahas et al., 2022; Kamzolov et al., 2023; Scieur, 2024). Despite all of the interest, even nowadays, many classical Quasi-Newton methods still lack non-asymptotic global convergence guarantees. Only recently global non-asymptotic convergence guarantees with explicit rates were established for BFGS in the strongly convex setting for specific line search procedures: Jin et al. (2024a) established rates for exact greedy line search and Jin et al. (2024b) established rates for Frank-Wolfe-type Armijo rules. Beyond classical Quasi-Newton methods, it is possible to prove global convergence rate by enhancing the update with cubic regularization, resulting in convergence guarantees in the convex case Kamzolov et al. (2023); Scieur (2024); Wang et al. (2024). However, those methods result in implicit update formula requiring additional line search in each iteration, involving matrix inversions (e.g., using the Woodbury identity (Woodbury, 1949; 1950)).

In this work, we aim to address all these challenges simultaneously – we aim to guarantee global non-asymptotic convergence guarantees for classical Quasi-Newton methods for non-strongly convex functions. To this end, we propose a simple stepsize schedule for the generic Quasi-Newton update (2) with guaranteed non-asymptotic global convergence in the convex setting. Our schedule is inspired by stepsize strategies developed for Damped Newton methods (Nesterov and Nemirovski, 1994; Hanzely et al., 2022; 2024), Cubic Regularized Newton methods (Nesterov and Polyak, 2006; Nesterov, 2008), and their inexact variants (Ghadimi et al., 2017; Agafonov et al., 2024a; 2023), as well as Cubic Regularized Quasi-Newton methods (Kamzolov et al., 2023; Scieur, 2024; Wang et al., 2024).

## 1.1 Contributions

- **From cubic regularization to explicit stepsize schedules:** We propose a simple stepsize schedule derived from the cubically regularized Quasi-Newton method, which we call *Cubically Enhanced Quasi-Newton* (CEQN) method. We obtain the schedule by carefully selecting the norm of the cubic regularization.
- **Global convergence guarantees:** We provide a convergence analysis for general convex functions. Under the assumption that Hessian approximations satisfy a relative inexactness condition, $(1 - \underline{\alpha})\mathbf{B}_x \preceq \nabla^2 f(x) \preceq (1 + \overline{\alpha})\mathbf{B}_x$ with $0 \leq \underline{\alpha} \leq 1$, $0 \leq \overline{\alpha}$, we prove the global rate $O\left(\frac{(\underline{\alpha}+\overline{\alpha})D^2}{K} + \frac{(1+\overline{\alpha})^{3/2}LD^3}{K^2}\right)$.
- **Adaptiveness:** We introduce an adaptive stepsize variant that automatically adjusts to the local accuracy of the Hessian approximation. The method naturally adapts to the local curvature without requiring stepsizes tuning and achieves $\varepsilon$-accuracy in $O\left(\frac{\alpha\overline{D}}{\varepsilon} + \frac{(1+\alpha)^{3/2}L\overline{D}^3}{\sqrt{\varepsilon}}\right)$ iterations, where $\alpha = \max(\underline{\alpha}, \overline{\alpha})$.
- **Verifiable criterion for inexactness.** We provide an implementable criterion for controlling Hessian inexactness that guarantees a global convergence rate of $\mathcal{O}(1/k^2)$ when the inexactness can be adaptively adjusted. This applies, for example, to Quasi-Newton methods with sampled curvature pairs or to stochastic second-order methods.
- **Experimental comparison.** We demonstrate that CEQN stepsizes, when combined with adaptive schemes for adjusting inexactness levels, consistently outperform standard Quasi-Newton methods and Quasi-Newton updates with fixed cubic regularization—both in terms of iteration count and wall-clock time.

## 1.2 Notation

We denote the global minimizer of the objective function $f$ (1) by $x_*$. The Euclidean norm is denoted by $\|\cdot\|$. We will use norms based on a symmetric positive definite matrix $\mathbf{B} \in \mathbb{R}^{d \times d}$ its inverse $\mathbf{H} \stackrel{\text{def}}{=} \mathbf{B}^{-1}$. For all $x, g \in \mathbb{R}^d$,

$$\|h\|_{\mathbf{B}} \stackrel{\text{def}}{=} \langle h, \mathbf{B}h \rangle^{1/2} = \langle h, \mathbf{H}^{-1}h \rangle^{1/2} \stackrel{\text{def}}{=} \|h\|_{\mathbf{H}}^*, \quad \|g\|_{\mathbf{B}}^* \stackrel{\text{def}}{=} \langle g, \mathbf{B}^{-1}g \rangle^{1/2} = \langle g, \mathbf{H}g \rangle^{1/2} \stackrel{\text{def}}{=} \|g\|_{\mathbf{H}}.$$

We denote Hessian and its inverse approximations at point $x$ as $\mathbf{B}_x$ and $\mathbf{H}_x$. If the approximation is evaluated at the point $x_k$, the $k$-th iterate of some algorithm, we write $\mathbf{B}_k \stackrel{\text{def}}{=} \mathbf{B}_{x_k}$ and $\mathbf{H}_k \stackrel{\text{def}}{=} \mathbf{H}_{x_k}$.

Notably, for updates $x_{k+1} = x_k - \eta_k \mathbf{H}_k \nabla f(x_k)$ it holds that $\|x_{k+1} - x_k\|_{\mathbf{B}_k} = \eta_k \|\nabla f(x_k)\|_{\mathbf{B}_k}^*$. We also define Hessian-induced norms

$$\|h\|_x \stackrel{\text{def}}{=} \left\langle h, \nabla^2 f(x) h \right\rangle^{1/2}, \quad \|g\|_x^* \stackrel{\text{def}}{=} \left\langle g, \nabla^2 f(x)^{-1} g \right\rangle^{1/2}.$$

For iterates $x_k,\ k \geq 0$ we denote $\|h\|_{x_k} \stackrel{\text{def}}{=} \|h\|_k$, and $\|g\|_{x_k}^* \stackrel{\text{def}}{=} \|g\|_k^*$. We define the operator norm with respect to the local Hessian norm as

$$\|\mathbf{A}\|_{op} \stackrel{\text{def}}{=} \sup_{y \in \mathbb{R}^d} \frac{\|\mathbf{A}y\|_x^*}{\|y\|_x}.$$

## 2 REGULARIZATION PERSPECTIVE ON QUASI-NEWTON METHODS

In this section, we motivate our stepsizes via a regularization perspective. Quasi-Newton methods can be seen as an approximation of the classical Newton method update, which at iterate $x$ can be written as the minimizer of the second-order Taylor approximation,

$$Q_f(y; x) \stackrel{\text{def}}{=} f(x) + \langle \nabla f(x), y - x \rangle + \tfrac{1}{2} \left\langle \nabla^2 f(x)(y - x), y - x \right\rangle. \tag{3}$$

Since the exact Hessian $\nabla^2 f(x)$ is typically unavailable or expensive to compute, Quasi-Newton methods replace it with a positive-definite approximation $\mathbf{B}_x \approx \nabla^2 f(x)$. This yields an inexact second-order model:

$$\overline{Q}_f(y; x) \stackrel{\text{def}}{=} f(x) + \langle \nabla f(x), y - x \rangle + \tfrac{1}{2} \left\langle \mathbf{B}_x(y - x), y - x \right\rangle, \tag{4}$$

which is minimized in classical Quasi-Newton methods, leading to the update (2) with $\eta_k = 1$.

Models (3) and (4) serve as local approximations of the objective function. Their accuracy can be quantified in terms of the smoothness of the Hessian and the quality of the Hessian approximation. If the Hessian of $f$ is $L_2$-Lipschitz continuous (i.e., $\|\nabla^2 f(x) - \nabla^2 f(y)\| \leq L_2 \|x - y\|$ for all $x, y \in \mathbb{R}^2$), then inexactness of Newton model can be bound as (Nesterov and Polyak, 2006):

$$|f(y) - Q_f(y; x)| \leq \tfrac{L_2}{6} \|x - y\|^3, \quad \forall x, y \in \mathbb{R}^d.$$

Bounding the inexactness of the Quasi-Newton model requires an additional assumption on the quality of the Hessian approximation $\|\mathbf{B}_x - \nabla^2 f(x)\| \leq \delta$. Then it holds (Agafonov et al., 2024a)

$$|f(y) - \overline{Q}_f(y; x)| \leq \tfrac{L_2}{6} \|x - y\|^3 + \delta \|x - y\|^2, \quad \forall x, y \in \mathbb{R}^d$$

These bounds demonstrate that the Taylor models are accurate in a neighborhood of $x$ as long as the curvature of function is smooth and the Hessian approximation $\mathbf{B}_x$ remains close to the true Hessian. Unfortunately, this guarantees the convergence only locally. In fact, both Newton's method and Quasi-Newton methods can diverge if initialized far from the solution (Jarre and Toint, 2016; Mascarenhas, 2007). This is because these models (3) and (4) do not provide a global upper bound on the function $f$, and may significantly underestimate it far from the current iterate.

One way to ensure the global convergence in (Quasi-)Newton methods is to introduce a stepsize schedule $\eta_k$ into the update. This modification can be naturally interpreted through the lens of regularization. In particular, the Quasi-Newton update (2) can be rewritten as

$$x_{k+1} = x_k - \eta_k \mathbf{H}_k \nabla f(x_k) = \operatorname{argmin}_{x \in \mathbb{R}^d} \left\{ f(x_k) + \langle \nabla f(x_k), x - x_k \rangle + \tfrac{1}{2\eta_k} \|x - x_k\|_{\mathbf{B}_k}^2 \right\}.$$

This viewpoint also highlights a key geometric property: if the stepsize $\eta_k$ and norm $\|\cdot\|_{\mathbf{B}_k}$ are affine-invariant[2], then the Quasi-Newton method itself is affine-invariant, which aligns with the common knowledge (Lyness, 1979). Affine-invariance property is practically significant, as it implies invariance to scaling and choice of the coordinate system, facilitating the implementation of the algorithm. Preserving it throughout the proofs requires careful technical analysis.

Another globalization strategy for the approximations (3) and (4) is to enhance them with a cubic regularization term Nesterov and Polyak (2006); Ghadimi et al. (2017). The Cubic Regularized Newton step takes the form

$$x_{k+1} = \operatorname{argmin}_{x \in \mathbb{R}^d} \left\{ Q_f(x; x_k) + \tfrac{L_2}{3} \|x - x_k\|^3 \right\}.$$

---

[2] Affine-invariance is invariance to affine transformations $f \to A \circ f$ for any linear operator $A$.

For functions with $L_2$-Lipschitz Hessian, the cubic model provides a global upper bound: $f(y) \leq Q_f(y; x) + \frac{L_2}{3}\|y - x\|^3$ for any $x, y \in \mathbb{R}^d$. In the case of inexact Hessians, additional regularization is required to restore the upper-bounding property Agafonov et al. (2024a). Specifically, the cubic regularized step becomes

$$x_{k+1} = \text{argmin}_{x \in \mathbb{R}^d} \left\{ \overline{Q}_f(x; x_k) + \frac{\delta}{2}\|x - x_k\|^2 + \frac{L_2}{3}\|x - x_k\|^3 \right\}$$

under which the objective is bounded above as $f(y) \leq \overline{Q}_f(y; x) + \frac{\delta}{2}\|y - x\|^2 + \frac{L_2}{3}\|y - x\|^3$ for any $x, y \in \mathbb{R}^d$ Agafonov et al. (2024a).

While cubic regularization enables global convergence guarantees, we highlight two limitations of the approach. First, the resulting step can be equivalently written as $x_{k+1} = x_k - (\nabla^2 f(x_k) + \lambda_k I)^{-1}\nabla f(x_k)$, with implicit $\lambda_k = L_2\|x_k - x_{k+1}\|$. Since $\lambda_k$ depends on the unknown next iterate $x_{k+1}$, it requires using an additional subroutine for solving the subproblem each iteration (Nesterov, 2021b). Secondly, usage of the non-affine-invariant Euclidean norm removes the desired affine-invariant property.

To address the loss of affine-invariance problems, Hanzely et al. (2022) adjusted the geometry of cubic regularization from Euclidean norm to local norm $\|\cdot\|_x^3$; matching the norm of quadratic term of Taylor polynomial. This resulted in the update preserving Newton direction with an adjusted stepsize.

## 2.1 CUBICALLY-ENHANCED QUASI-NEWTON

Leveraging these ideas, we propose a regularization strategy for Quasi-Newton methods that aligns quadratic and cubic terms in the same geometry using norms $\|\cdot\|_{\mathbf{B}_k}$. This preserves the update direction of the classical Quasi-Newton methods, enhanced with a stepsize reflecting both curvature and model accuracy, hence we call it *Cubically-Enhanced Quasi-Newton* (CEQN). Notably, it enjoys the structure of Quasi-Newton steps and global convergence guarantees of cubic regularized methods. Let us formalize the mentioned claims. CEQN method minimizes the regularized model,

$$x_{k+1} \stackrel{\text{def}}{=} \text{argmin}_{y \in \mathbb{R}^d} \left\{ f(x_k) + \langle \nabla f(x_k), y - x_k \rangle + \frac{\theta}{2}\|y - x_k\|_{\mathbf{B}_k}^2 + \frac{L}{3}\|y - x_k\|_{\mathbf{B}_k}^3 \right\}, \quad (5)$$

which we simplify using notation $h_k \stackrel{\text{def}}{=} x_{k+1} - x_k$. The first-order optimality condition yields

$$0 = \nabla f(x_k) + \theta \mathbf{B}_k h_k + L\|h_k\|_{\mathbf{B}_k}\mathbf{B}_k h_k, \quad (6)$$

which we multiply by $\mathbf{B}_k^{-1}$ and rearrange, obtaining

$$h_k = -\left(\theta + L\|h_k\|_{\mathbf{B}_k}\right)^{-1}\mathbf{B}_k^{-1}\nabla f(x_k).$$

This shows that the update direction matches classical Quasi-Newton methods; with a stepsize

$$\eta_k \stackrel{\text{def}}{=} \left(\theta + L\|h_k\|_{\mathbf{B}_k}\right)^{-1}.$$

Substituting back $h_k = -\eta_k \mathbf{B}_k^{-1}\nabla f(x_k)$ and $\|h_k\|_{\mathbf{B}_k} = \eta_k\|\nabla f(x_k)\|_{\mathbf{H}_k}$ into (6) simplifies the equation to $0 = \left(1 - \theta\eta_k + \eta_k^2 L\|\nabla f(x_k)\|_{\mathbf{B}_k}^*\right)\nabla f(x_k)$ and solving the quadratic equation in $\eta_k$ gives the closed-form expression:

$$\eta_k = \frac{2}{\theta + \sqrt{\theta^2 + L\|\nabla f(x_k)\|_{\mathbf{B}_k}^*}}. \quad (7)$$

Hence, the minimizer of (5) is algebraically identical to the classical Quasi-Newton update (2) with stepsize (7). In the special case of an exact Hessian approximation and $\theta = 1$, this method reduces to Affine-Invariant Cubic Newton method of Hanzely et al. (2022).

Quasi-Newton methods are considered to be inexact approximations of the Newton method. This result provides alternative interpretation, as exact minimizers of the Newton method in an adjusted geometry. To obtain convergence rates we need to bound the difference between those geometries.

Before presenting convergence rate guarantees, let us formally list CEQN as an Algorithm 1. We note that if the parameters $L$ and $\theta$ are chosen such that the initial stepsize $\eta_0$ from (7) matches the best fine-tuned constant learning rate of a given Quasi-Newton method, then enhancing it with CEQN stepsizes can lead to faster convergence. This is because the $\mathbf{B}$-norm of the gradient naturally decreases as the method approaches the solution.

---

**Algorithm 1** Cubically Enhanced Quasi-Newton Method

---

1: **Requires:** Initial point $x_0 \in \mathbb{R}^d$, constants $L, \theta > 0$.
2: **for** $k = 0, 1, \ldots, K$ **do**
3:     $\eta_k = \frac{2}{\theta + \sqrt{\theta^2 + L\|\nabla f(x_k)\|_{\mathbf{H}_k}}}$
4:     $x_{k+1} = x_k - \eta_k \mathbf{H}_k \nabla f(x_k)$
5: **Return:** $x_{K+1}$

---

## 3 CONVERGENCE RESULTS

As we mentioned before, CEQN is affine-invariant. If we aim to obtain affine-invariant convergence guarantees, we have to base our analysis on affine-invariant smoothness assumption. Throughout this work we consider the class of semi-strongly self-concordant functions introduced in Hanzely et al. (2022). This class is an affine-invariant version of second-order smoothness, and is positioned between standard self-concordance and strong self-concordance of Rodomanov and Nesterov (2021a),

$$\text{strong self-concordance} \subseteq \text{semi-strong self-concordance} \subseteq \text{self-concordance}.$$

**Assumption 1.** *Convex function $f \in C^2$ is called semi-strongly self-concordant if*

$$\left\| \nabla^2 f(y) - \nabla^2 f(x) \right\|_{op} \le L_{semi} \|y - x\|_x, \quad \forall y, x \in \mathbb{R}^d. \tag{8}$$

Semi-strong self-concordance yields explicit second-order approximation bounds on both the function and its gradient (Hanzely et al., 2022), for all $x, y \in \mathbb{R}^d$, we have:

$$|f(y) - Q_f(y;x)| \le \tfrac{L_{\text{semi}}}{6} \|y - x\|_x^3, \quad \left\| \nabla f(y) - \nabla f(x) - \nabla^2 f(x)(y - x) \right\|_x^* \le \tfrac{L_{\text{semi}}}{2} \|y - x\|_x^2.$$

We now introduce a relative inexactness condition that quantifies how closely the approximate Hessian $\mathbf{B}_x$ tracks the true Hessian $\nabla^2 f(x)$.

**Assumption 2.** *For a function $f(x)$ and point $x \in \mathbb{R}^d$, a positive definite matrix $B_x \in \mathbb{R}^{d \times d}$ is considered a $(\underline{\alpha}, \overline{\alpha})$-relative inexact Hessian with $0 \le \underline{\alpha} \le 1$, $0 \le \overline{\alpha}$ if it satisfies the inequality*

$$(1 - \underline{\alpha})\mathbf{B}_x \preceq \nabla^2 f(x) \preceq (1 + \overline{\alpha})\mathbf{B}_x. \tag{9}$$

Combining Assumptions 1 and 2, we obtain the following estimates comparing the function $f(y)$, its gradient $\nabla f(y)$, and their inexact second-order model $\overline{Q}(y;x)$ (4)

**Lemma 1.** *Let Assumptions 1 and 2 hold. Then, for any $x, y \in \mathbb{R}^d$, the following inequalities hold:*

$$f(y) - \overline{Q}_f(y;x) \le \tfrac{\overline{\alpha}}{2} \|y - x\|_{\mathbf{B}_x}^2 + \tfrac{(1+\overline{\alpha})^{3/2} L_{\text{semi}}}{6} \|y - x\|_{\mathbf{B}_x}^3, \tag{10}$$

$$\overline{Q}_f(y;x) - f(y) \le \tfrac{\underline{\alpha}}{2} \|y - x\|_{\mathbf{B}_x}^2 + \tfrac{(1+\overline{\alpha})^{3/2} L_{\text{semi}}}{6} \|y - x\|_{\mathbf{B}_x}^3, \tag{11}$$

$$\|\nabla \overline{Q}_f(y;x) - \nabla f(y)\|_{\mathbf{B}_x}^* \le \alpha_{\max} \|y - x\|_{\mathbf{B}_x} + \tfrac{(1+\overline{\alpha})^{3/2} L_{\text{semi}}}{2} \|y - x\|_{\mathbf{B}_x}^2, \tag{12}$$

*where $\alpha_{\max} := \max(\underline{\alpha}, \overline{\alpha})$.*

**Theorem 1.** *Let Assumptions 1, 2 hold, $f$ be a convex function, and*

$$D \stackrel{def}{=} \max_{k \in [0; K+1]} \|x_k - x_*\|_{\mathbf{B}_k}.$$

*After $K + 1$ iterations of Algorithm 1 with parameters*

$$\theta \ge 1 + \overline{\alpha}, \ L \ge \tfrac{(1+\overline{\alpha})^{3/2} L_{semi}}{2},$$

*we get the following bound*

$$f(x_{K+1}) - f(x_*) \le \tfrac{(\underline{\alpha}+\overline{\alpha})}{2} \tfrac{9D^2}{K+3} + (1 + \overline{\alpha})^{3/2} \tfrac{3 L_{semi} D^3}{(K+1)(K+2)}.$$

This result provides an explicit upper bound on the objective residual after $K + 1$ iterations of Algorithm 1. The second term on the right-hand side matches the convergence rate of the Cubic

Regularized Newton method Nesterov (2008) and accelerated gradient descent Nesterov (1983), and reflects the ideal behavior under exact second-order information. The first term accounts for the effect of Hessian inexactness and aligns with the standard convergence rate of gradient descent. However, when the inexactness can be explicitly controlled – e.g., by increasing the batch size in stochastic settings or refining the approximation scheme – the convergence rate can closely match that of the exact Cubic Regularized method. To formalize the conditions required to achieve this rate and provide further insight into the performance of CEQN, we introduce the following lemma.

**Lemma 2.** *Let Assumptions 1, 2 hold and $f(x)$ be a convex function. Algorithm 1 with parameters $\theta = 1 + \alpha \geq 1 + \alpha_{max}$, $L \geq (1+\overline{\alpha})^{3/2}L_{semi}$ implies the following one-step decrease*

$$f(x_k) - f(x_{k+1}) \geq \min\left\{ \left(\tfrac{1}{4\alpha}\right)\left(\|\nabla f(x_{k+1})\|^*_{\mathbf{B}_k}\right)^2, \left(\tfrac{1}{6L}\right)^{\frac{1}{2}}\left(\|\nabla f(x_{k+1})\|^*_{\mathbf{B}_k}\right)^{3/2}\right\} \geq 0. \quad (13)$$

**Remark 1.** *Let Assumptions 3, 4 hold and let $\underline{\alpha} < 1$. Assume that the level set of $f$ is bounded:*

$$\max_{x \in \mathcal{L}(x_0)} \|x - x_*\| \leq R < \infty, \text{ where } \mathcal{L}(x_0) = \{x \mid f(x) \leq f(x_0)\}. \quad (14)$$

*Then $D$ depends only on the constants $\underline{\alpha}$, $R$, $L_{semi}$, $\|\nabla^2 f(x_*)\|$.*

An immediate corollary of this lemma is that Algorithm 1 generates a monotonically non-increasing sequence of function values, with a strict decrease whenever $\nabla f(x_{k+1}) \neq 0$. The lemma also implies that CEQN transitions only once between two convergence regimes, determined by which term in the minimum on the right-hand side of (13) is active. As a result, CEQN initially benefits from the faster convergence rate characteristic of the Cubic Regularized Newton method.

**Corollary 1.** *Let Assumptions 1, 2 hold and $f$ be a convex function. Algorithm 1 with parameters $\theta = 1 + \alpha \geq 1 + \alpha_{max}$, $L \geq (1+\overline{\alpha})^{3/2}L_{semi}$ converges with the rate $\mathcal{O}(k^{-2})$ until it reaches the region $\|\nabla f(x_{k+1})\|^*_{\mathbf{B}_k} \leq \frac{4\alpha^2}{9L^2(1+\alpha)^3}$.*

And finally, the following corollary of Lemma 2 provides a sufficient condition on the inexactness levels $\alpha_k$ to maintain the global convergence rate $\mathcal{O}(1/k^2)$.

**Corollary 2.** *Let Assumptions 1 and 2 hold, and let $f$ be a convex function. Suppose Algorithm 1 is run with parameters $\theta_k = 1 + \alpha_k \geq 1 + \alpha_{\max}$ and $L \geq (1+\overline{\alpha})^{3/2}L_{semi}$. If the inexactness satisfies $\alpha_k \leq L\|x_{k+1}-x_k\|_{\mathbf{B}_k}$, then Algorithm 1 achieves the convergence rate $f(x_{k+1})-f(x^*) = \mathcal{O}(k^{-2})$.*

Note that the inexactness condition is verifiable in practice, indicating that the method can adapt its behavior in scenarios where the inexactness is controllable.

# 4    ADAPTIVE SCHEME

In this section, we present a modification of CEQN that automatically adapts to the level of inexactness in the Hessian approximation. Our adaptive method, Algorithm 2, incrementally increases the inexactness parameter $\alpha_k$ until the model decrease condition is satisfied.

---
**Algorithm 2** Adaptive Cubically Enhanced Quasi-Newton Method (backtracking acceptance)

---
1: **Requires:** Initial point $x_0 \in \mathbb{R}^d$, constants $L, \alpha_0 > 0$, increase multiplier $\gamma_{inc} > 1$.
2: **for** $k = 0, 1, \ldots, K$ **do**
3:     $\eta_k = \frac{2}{(1+\alpha_k)+\sqrt{(1+\alpha_k)^2+(1+\alpha_k)^{3/2}L\|\nabla f(x_k)\|^*_{\mathbf{B}_k}}}$
4:     $x_{k+1} = x_k - \eta_k \mathbf{H}_k \nabla f(x_k)$
5:     **while** $\langle \nabla f(x_{t+1}), x_k - x_{k+1}\rangle \leq \min\left\{ \frac{\left(\|\nabla f(x_{k+1})\|^*_{\mathbf{B}_k}\right)^2}{4\alpha_k}, \frac{\left(\|\nabla f(x_{t+1})\|^*_{\mathbf{B}_k}\right)^{3/2}}{(6(1+\alpha_k)^{3/2}L)^{1/2}}\right\}$ **do**
6:         $\alpha_k = \alpha_k \gamma_{inc}$
7:         Recompute $\eta_k$ as in Line 3
8:         Update $x_{k+1}$ as in Line 4
9: $\alpha_{k+1} = \alpha_k$
10: **Return:** $x_{K+1}$

---

**Theorem 2.** *Let Assumptions 1, 2 hold, $f$ be a convex function, $\varepsilon > 0$ be the target accuracy, and*

$$\overline{D} \overset{def}{=} \max_{k \in [0;K+1]} \left( \|x_k - x_*\|_{\mathbf{B}_k} + \|\nabla f(x_k)\|^*_{\mathbf{B}_k} \right). \tag{15}$$

*Suppose Algorithm 2 is run with parameters $L \geq 2L_{semi}$, $\alpha_0 > 0$, $\gamma_{inc} > 1$. Then, to obtain a point $x_K$ such that $f(x_K) - f(x^*) \leq \varepsilon$, it suffices to perform $K$ iterations of Algorithm 2 for*

$$K = \mathcal{O}\left( \frac{\alpha_K \overline{D}^2}{\varepsilon} + \frac{(1+\alpha_K)^{3/2} L \overline{D}^3}{\sqrt{\varepsilon}} + \log_{\gamma_{inc}}\left( \frac{\alpha_K}{\alpha_0} \right) \right). \tag{16}$$

**Remark 2.** *Let Assumptions 3, 4 hold and let $\underline{\alpha} < 1$. Assume that the level set of $f$ is bounded* (14). *Then $\overline{D}$ depends only on the constants $\underline{\alpha}$, $R$, $\overline{L}_{semi}$, $\|\nabla^2 f(x_*)\|$.*

The convergence rate (16) consists of three components: the first term reflects the effect of inexactness in the Hessian approximation and corresponds to the gradient descent rate; the second term matches the convergence rate of the exact Cubic Regularized Newton method; and the third term accounts for the additional iterations incurred by the inexactness correction procedure.

All supplementary results established for Algorithm 1 extend to the adaptive version as well. In particular:

- the one-step decrease and monotonicity properties (Lemma 2) remain valid,
- the transition between cubic and gradient convergence phases still occurs only once (as in Corollary 1, with $\alpha \to \alpha_k$),
- and the sufficient condition for achieving the global $\mathcal{O}(k^{-2})$ rate under controllable inexactness remains unchanged (Corollary 2).

## 5 PRACTICAL PERFORMANCE

In this section, we evaluate the practical performance of the proposed CEQN stepsizes. We begin by discussing the practicality and implementability of the proposed methods.

CEQN stepsize (Algorithm 1) relies on two hyperparameters: the cubic regularization parameter $L$ and the quadratic regularization parameter $\theta$. To reduce the burden of tuning and enhance usability, we introduced an adaptive Algotrithm 2 in the previous section, which replaces the two parameters with a single adaptive sequence $\alpha_k$. This sequence is intended to track the level of approximation error in the Hessian model.

However, the original adaptive scheme suffers from a notable limitation: it only allows $\alpha_k$ to increase throughout the optimization process. As a result, the algorithm tends to significantly overestimate the actual inexactness level, which in turn degrades performance. This design choice was made to ensure the validity of theoretical convergence guarantees—allowing $\alpha_k$ to decrease would make it difficult to control the number of inexactness correction steps, thus breaking the proof structure.

**Practical Modifications.** To address this issue, we propose two practical variants of the Adaptive CEQN stepsize strategy. Both variants use a monotonic decay scheme in which the inexactness level $\alpha_k$ is multiplicatively decreased after each successful step, allowing the optimizer to better adapt to local curvature. The only difference between them lies in the condition used to decide whether a step is successful.

The first variant which we denote the `dual` condition uses the theoretical regularity condition from our analysis (Line 5 of Algorithm 2) and leads to one step decrease shown in Lemma 2. The second variant, denoted `reg` condition adopts a similar condition to Adaptive Cubic Regularized Quasi-Newton. Its ensures that the next iterate satisfies sufficient decrease condition `CheckAccept`:

$$f(x_{k+1}) \leq f(x_k) - \frac{1}{2}\eta_k \left( \|\nabla f(x_k)\|^*_{\mathbf{B}_k} \right)^2 - \frac{L}{6}\eta_k^3 \left( \|\nabla f(x_k)\|^*_{\mathbf{B}_k} \right)^3 \tag{17}$$

with $\theta = 1 + \alpha_k$, $L = L_{semi}(1 + \alpha_k)^{3/2}$. This is supported by the following result:

**Lemma 3.** *Let Assumptions 1, 2 hold. Step* (2) *with CEQN stepsize* (7) *and with parameters $\theta \geq 1 + \alpha_{max}$, $L \geq (1 + \overline{\alpha})^{3/2} L_{semi}$ implies one-step decrease* (17).

A theoretical bound of this form can be found in (Nesterov and Polyak, 2006, Lemma 4), where it is used in the analysis of the Cubic Regularized Newton method for the nonconvex case. Although we cannot guarantee a global iteration complexity bound for Algorithm 3, both conditions ensure a provable decrease in the objective at each step.

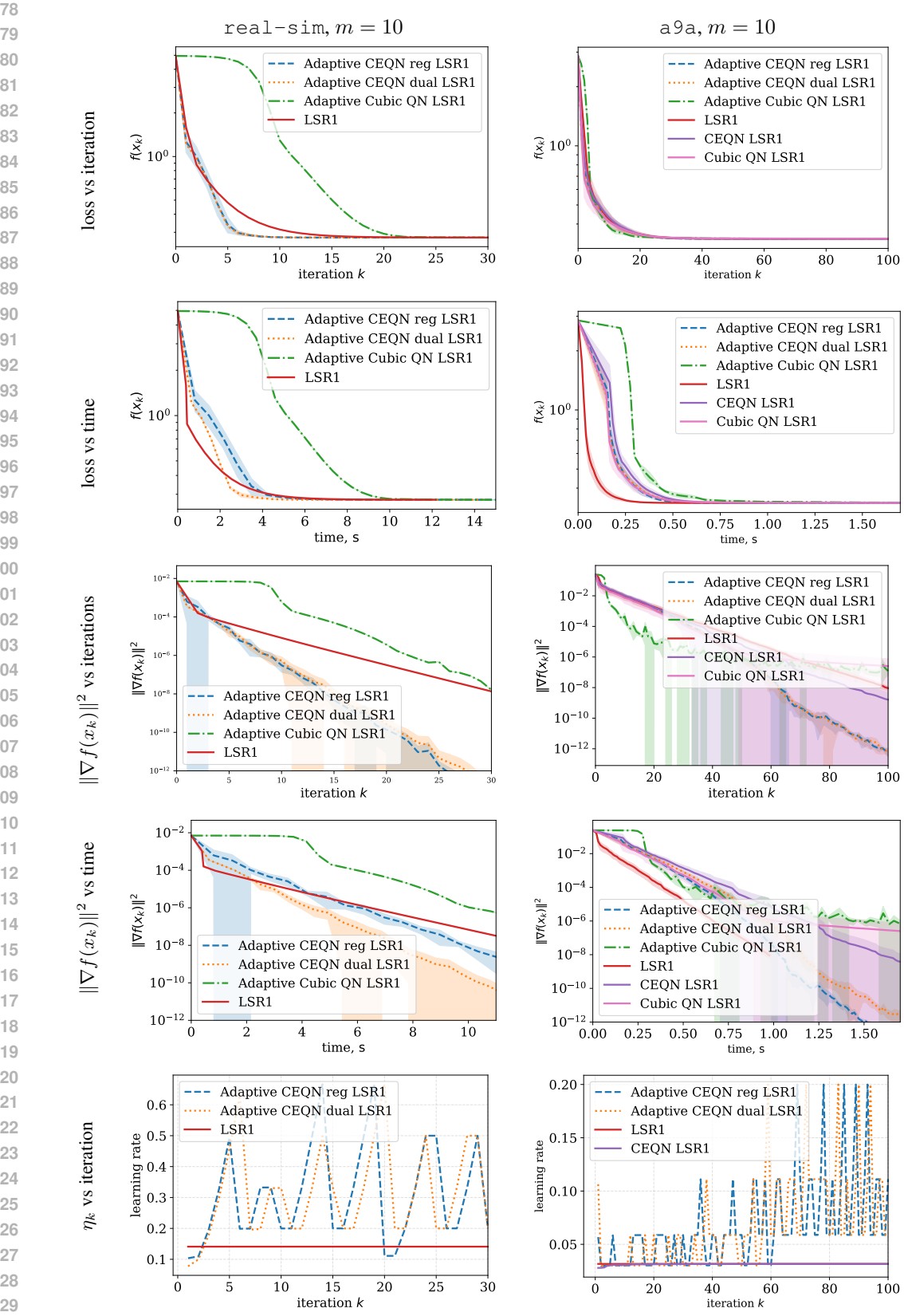

Figure 1: Performance on `a9a` and `real-sim` datasets.

---

**Algorithm 3** Practical Adaptive Cubically Enhanced Quasi-Newton Method (backtracking acceptance)

---

1: **Requires:** Initial point $x_0 \in \mathbb{R}^d$, constants $L, \alpha_0 > 0$, increase multiplier $\gamma_{inc} > 1$, decrease multiplier $0 < \gamma_{\text{dec}} < 1$, mode $\in \{\texttt{reg}, \texttt{dual}\}$.
2: **for** $k = 0, 1, \ldots, K$ **do**
3:     $\eta_k = \frac{2}{(1+\alpha_k) + \sqrt{(1+\alpha_k)^2 + (1+\alpha_k)^{3/2} L \|\nabla f(x_k)\|^*_{\mathbf{B}_k}}}$
4:     $x_{k+1} = x_k - \eta_k \mathbf{H}_k \nabla f(x_k)$
5:     **while not** $\texttt{CheckAccept}(x_{k+1}, \texttt{mode})$ **do**
6:        $\alpha_k = \alpha_k \gamma_{inc}$
7:        Recompute $\eta_k$ as in Line 3
8:        Update $x_{k+1}$ as in Line 4
9:     $\alpha_{k+1} = \alpha_k \gamma_{\text{dec}}$
10: **Return:** $x_{K+1}$

---

**Hessian Approximation.** Experiments presented in this section approximate the inverse Hessian $\mathbf{H}_k \approx \nabla^2 f(x_k)$ using limited-memory SR1 method based on $m$ sampled curvature pairs $(s_i, y_i)$. These pairs are generated by sampling random directions $d_i \sim \mathcal{N}(0, I)$ and computing s $s_i = d_i, y_i = \nabla^2 f(x_k) d_i$ via Hessian-vector product. This sampling-based approach decouples curvature estimation from the optimization trajectory and may offer improved robustness. We set the initial inverse Hessian approximation as $\mathbf{H}_k^0 = \mathbf{H}_0 = cI$ with $c > 0$ and compute the product $\mathbf{H}_k \nabla f(x_k)$ using the limited-memory SR1 update in a compact recursive form:

$$\mathbf{H}_k^{i+1} \nabla f(x_k) = \mathbf{H}_k^i \nabla f(x_k) + \frac{(s_i - \mathbf{H}_k^i y_i)^T \nabla f(x_k)}{(s_i - \mathbf{H}_k^i y_i)^T y_i}(s_i - \mathbf{H}_k^i y_i), \quad i \in [0, m].$$

where $\mathbf{H}_k = \mathbf{H}_k^m$ denotes the final approximation used at iteration $k$.

**Experimental Setup.** In this section we consider $l_2$ regularized logistic regression problem,

$$f(x) = \frac{1}{n}\sum_{i=1}^n \log(1 + \exp(-b_i a_i^\top x)) + \frac{\mu}{2}\|x\|^2, \tag{18}$$

where $(a_i, b_i)_{i=1}^n$ are training examples, with $a_i \in \mathbb{R}^d$ representing feature vectors and $b_i \in \{-1, 1\}$ the corresponding class labels. The parameter $\mu \geq 0$ controls the strength of $\ell_2$ regularization. We set $\mu = 10^{-4}$, and initialize the approximation as $10^{-4}I$, and set starting point as all-one vector. We use datasets from the $\texttt{LIBSVM}$ (Chang and Lin, 2011) collection: $\texttt{a9a}$ ($d = 123$) and $\texttt{real-sim}$ ($d = 20{,}958$) to evaluate performance. For the consistency, all experiments on a given dataset were conducted using the same workstation with $\texttt{NVIDIA RTX A6000}$.

We compare six algorithms on problem (18): LSR1, LSR1 with CEQN stepsizes (Algorithm 1), two versions of LSR1 with adaptive stepsizes (Algorithm 3), and Cubic Regularized Quasi-Newton with LSR1 updates (Kamzolov et al., 2023), both with and without adaptivity.

We fine-tune all hyperparameters via grid searches. For LSR1, we tune the parameter $L$ and use stepsize $\eta_k = 1/L$. For Algorithm 1, we tune both $\theta = 1 + \alpha$ and $L$. For Algorithm 3, we fix $\alpha_0 = 1$ and tune $L$. For Cubic Quasi-Newton, we tune $(L, \delta)$ in the non-adaptive case, and $L$ in the adaptive variant, where we set $\delta_0 = 0.1$. All adaptive algorithms use $\gamma_{\text{inc}} = 2$ and $\gamma_{\text{dec}} = 0.5$. All algorithms are run with $m = 10$ and evaluated across 5 different random seeds. The complete hyper-parameter search grids, the best-tuned values, and further experimental results are reported in the Appendix.

**Results.** We present convergence results on Figure 1. On the larger $\texttt{real-sim}$ dataset—where we compared only the adaptive variants and standard LSR1—the benefit of the proposed CEQN stepsize is pronounced. CEQN consistently outperforms the competing algorithms in both iteration count and wall-clock time when measured by log-loss and gradient-squared. A key insight is provided by the step-size evolution plot: the adaptive schemes automatically adjust to the accuracy of the Hessian approximation, allowing their steps to grow well beyond the fixed, optimal step length used by classical LSR1. A similar pattern is observed on the $\texttt{a9a}$ dataset. Although the difference in objective values is less visually striking, the increasing step sizes translate into faster gradient convergence. Finally, the loss- and gradient-squared-versus-time curves show that the number of extra inner updates required to satisfy the acceptance tests of Algorithm 3 is small. Consequently, CEQN achieves superior performance not only in terms of iterations but also in wall-clock time.

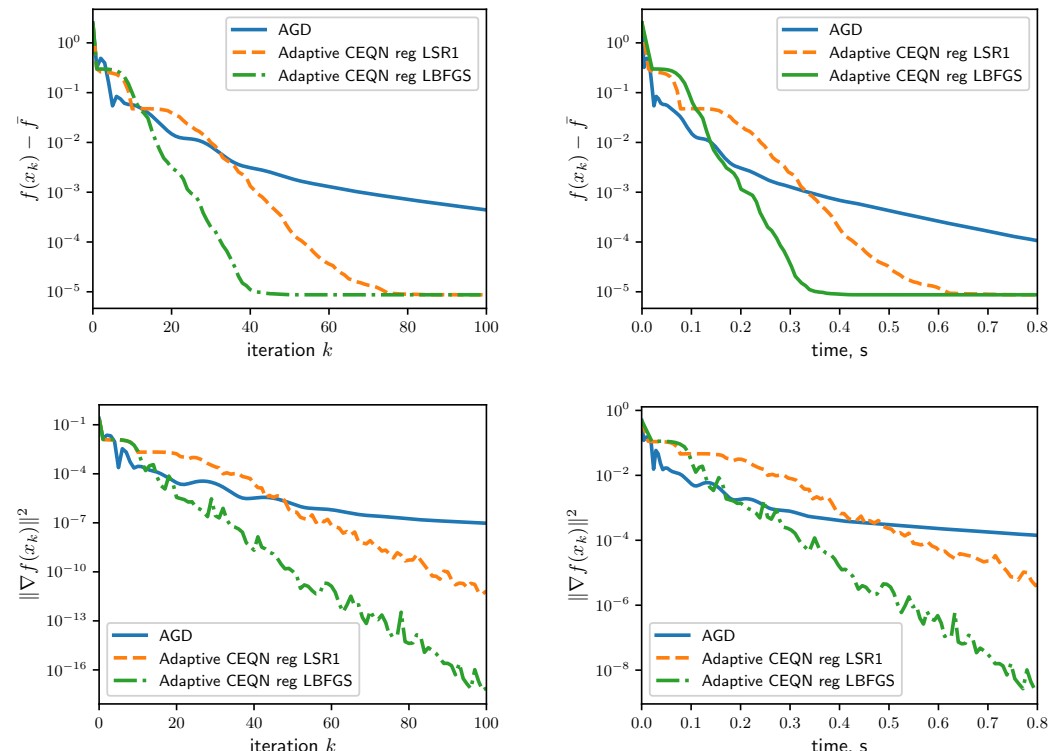

Figure 2: Comparison of AGD and CEQN with memory size $m = 10$ on `a9a`.

**Comparision with Accelerated Gradient Descent.** We additionally compare CEQN with Nesterov's Accelerated Gradient Descent (AGD). The experiment is conducted on the logistic regression problem (18) using the `a9a` dataset with regularization parameter $\mu = 10^{-4}$ and an all-ones initialization. For AGD, we tune two hyperparameters: the inverse stepsize $L$ over the same logarithmic grid as used for other methods (Appendix D.1), and the acceleration parameter $\beta \in 0.1, 0.2, 0.4, 0.5, 0.6, 0.7, 0.8, 0.9, 0.99$. The optimal configuration is $(L, \beta) = (0.1, 0.9)$.

For CEQN, we employ both LSR1 and L-BFGS Hessian approximations with the `reg` adaptive variant. Hyperparameters match those used in Figure 1 on `a9a`. The reference optimum $\bar{f}$ is computed using Newton's method. CEQN with both Hessian approximation strategies consistently outperforms AGD in terms of convergence per iteration and wall-clock time. While AGD appears faster within the suboptimality region $\lesssim 10^{-2}$, CEQN rapidly overtakes it and reaches suboptimality below $10^{-4}$ significantly sooner. This experiment was executed on a `MacBook Pro (Apple M2 Pro, 32GB RAM)`.

# 6 LIMITATIONS

This study focuses on Quasi-Newton (QN) methods equipped with an additional **B**-norm regularization that yields an explicit stepsize formula. The resulting stepsize, however, depends on two constants, one of which—the current accuracy level of the Hessian approximation—is unknown in practice. Although we mitigate this issue by proposing an adaptive strategy with provable convergence, the analysis guarantees adaptation only to the largest inaccuracy level encountered. For the more practical variant (Algorithm 3) we can prove only a one-step decrease; a full global convergence rate remains open. Intriguing directions for future research include whether a Hessian-approximation scheme can be devised that reaches the ideal $\mathcal{O}(1/k^2)$ rate without extra assumptions on inexactness, how strong-convexity parameters might reshape the CEQN stepsize and its guarantees, and whether an adaptive mechanism can be designed to track the current (rather than maximal) inexactness level while still retaining rigorous complexity bounds.

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

# Appendix

## A  OTHER RELATED WORKS

Second-order methods have a long and rich history, tracing back to the pioneering works (Newton, 1687; Raphson, 1697; Simpson, 1740; Bennett, 1916). Research in this area typically addresses two main aspects: local convergence properties and globalization strategies. For more historical context on the development of second-order methods, we refer to (Polyak, 2007).

A major breakthrough in globally convergent second-order methods came with the introduction of cubic regularization by Nesterov and Polyak (2006), who proposed augmenting the second-order Taylor approximation with a cubic term to guarantee global convergence, achieving a convergence rate matching that of accelerated gradient descent (Nesterov, 1983). This approach was further accelerated in (Nesterov, 2008), establishing a convergence rate that surpasses the lower bounds for first-order methods. These foundational works initiated a new line of research in second-order optimization, encompassing generalizations to higher-order derivatives (Baes, 2009; Nesterov, 2021b), near-optimal (Gasnikov et al., 2019; Bubeck et al., 2019) and optimal acceleration techniques (Kovalev and Gasnikov, 2022; Carmon et al., 2022), and faster convergence rates under higher smoothness assumptions (Nesterov, 2021c;a; Kamzolov, 2020; Doikov et al., 2024).

However, methods based on cubic or higher-order regularization typically require solving a nontrivial subproblem at each iteration, which introduces computational overhead. To mitigate this, several approaches have been proposed to simplify the cubic regularized Newton step, enabling explicit or efficiently computable updates (Polyak, 2009; 2017; Mishchenko, 2021; Doikov and Nesterov, 2021; Doikov et al., 2024; Hanzely et al., 2022). Such methods can also employ faster convergence under higher smoothness assumptions (Hanzely et al., 2024).

Even without cubic regularization, the classical Newton method is computationally demanding, as it requires solving a linear system involving the Hessian or computing its inverse at each iteration. Quasi-Newton methods (Dennis and Moré, 1977; Nocedal and Wright, 1999) address this by efficiently constructing low-rank approximations of the (inverse) Hessian, thereby reducing the per-iteration cost.

Another class of approaches reduces computational complexity by applying Newton-type updates in low-dimensional subspaces (Qu et al., 2016; Gower et al., 2019; Doikov and Richtárik, 2018; Hanzely et al., 2020), or by employing Hessian sketches to approximate curvature information (Pilanci and Wainwright, 2017; Xu et al., 2020; Kovalev et al., 2019). These techniques can also be integrated into cubic-regularized frameworks that admit explicit stepsizes (Hanzely, 2023).

Several works have investigated the impact of inexact Hessian information on the convergence behavior of cubic-regularized methods in standard optimization problems (Ghadimi et al., 2017; Agafonov et al., 2024a; Antonakopoulos et al., 2022; Agafonov et al., 2023), min-max optimization (Lin et al., 2022), and variational inequalities (Agafonov et al., 2024b). Additionally, second-order methods with inexact or stochastic derivatives have demonstrated strong performance in distributed optimization settings (Zhang and Lin, 2015; Daneshmand et al., 2021; Agafonov et al., 2021; Dvurechensky et al., 2022; Agafonov et al., 2022b;a).

More recently, efficient inexact second-order methods have been proposed specifically for large-scale training of language models (Gupta et al., 2018; Vyas et al., 2025; Liu et al., 2024; Jordan et al., 2024; Liu et al., 2025; Kovalev, 2025; Riabinin et al., 2025).

## B  CONVERGENCE ANALYSIS

Second-order Taylor approximation:

$$Q_f(y; x) \stackrel{\text{def}}{=} f(x) + \langle \nabla f(x), y - x \rangle + \tfrac{1}{2} \langle \nabla^2 f(x)(y - x), y - x \rangle. \tag{19}$$

Inexact second-order Taylor approximation:

$$\overline{Q}_f(y; x) \stackrel{\text{def}}{=} f(x) + \langle \nabla f(x), y - x \rangle + \tfrac{1}{2} \langle \mathbf{B}_x(y - x), y - x \rangle, \tag{20}$$

**Assumption 3.** *Convex function $f \in C^2$ is called semi-strongly self-concordant if*

$$\left\| \nabla^2 f(y) - \nabla^2 f(x) \right\|_{op} \leq L_{semi} \|y - x\|_x, \quad \forall y, x \in \mathbb{R}^d. \tag{21}$$

**Lemma 4** (Hanzely et al. (2022)). *If $f$ is semi-strongly self-concordant, then*

$$|f(y) - Q_f(y; x)| \leq \tfrac{L_{semi}}{6} \|y - x\|_x^3, \quad \forall x, y \in \mathbb{R}^d. \tag{22}$$

*Consequently, we have upper bound for function value in form*

$$f(y) \leq Q_f(y; x) + \tfrac{L_{semi}}{6} \|y - x\|_x^3. \tag{23}$$

**Lemma 5** (Hanzely et al. (2022)). *For semi-strongly self-concordant function $f$ holds*

$$\left\| \nabla f(y) - \nabla f(x) - \nabla^2 f(x)[y - x] \right\|_x^* \leq \frac{L_{semi}}{2} \|y - x\|_x^2. \tag{24}$$

**Assumption 4.** *For a function $f(x)$ and point $x \in \mathbb{R}^d$, a positive definite matrix $B_x \in \mathbb{R}^{d \times d}$ is considered a $(\underline{\alpha}, \overline{\alpha})$-relative inexact Hessian with $0 \leq \underline{\alpha} \leq 1$, $0 \leq \overline{\alpha}$ if it satisfies the inequality*

$$(1 - \underline{\alpha})\mathbf{B}_x \preceq \nabla^2 f(x) \preceq (1 + \overline{\alpha})\mathbf{B}_x, \tag{25}$$

**Lemma 6.** *Let Assumptions 3 and 4 hold. Then, for any $x, y \in \mathbb{R}^d$, the following inequalities hold:*

$$f(y) - \overline{Q}_f(y; x) \leq \tfrac{\overline{\alpha}}{2} \|y - x\|_{\mathbf{B}_x}^2 + \tfrac{(1+\overline{\alpha})^{3/2} L_{semi}}{6} \|y - x\|_{\mathbf{B}_x}^3, \tag{26}$$

$$\overline{Q}_f(y; x) - f(y) \leq \tfrac{\underline{\alpha}}{2} \|y - x\|_{\mathbf{B}_x}^2 + \tfrac{(1+\overline{\alpha})^{3/2} L_{semi}}{6} \|y - x\|_{\mathbf{B}_x}^3, \tag{27}$$

$$\|\nabla \overline{Q}_f(y; x) - \nabla f(y)\|_{\mathbf{B}_x}^* \leq \alpha_{\max} \|y - x\|_{\mathbf{B}_x} + \tfrac{(1+\overline{\alpha})^{3/2} L_{semi}}{2} \|y - x\|_{\mathbf{B}_x}^2, \tag{28}$$

*where $\alpha_{\max} := \max(\underline{\alpha}, \overline{\alpha})$.*

*Proof.* For any $x, y \in \mathbb{R}^d$,

$$\begin{aligned}
f(y) - \overline{Q}_f(y; x) &= f(y) - Q_f(y; x) + Q_f(y; x) - \overline{Q}_f(y; x) \\
&\overset{(23)}{\leq} \tfrac{L_{semi}}{6} \|y - x\|_x^3 + Q_f(y; x) - \overline{Q}_f(y; x) \\
&= \tfrac{L_{semi}}{6} \|y - x\|_x^3 + \tfrac{1}{2} \left\langle (\nabla^2 f(x) - \mathbf{B}_x)(y - x), (y - x) \right\rangle \\
&\overset{(25)}{\leq} \tfrac{L_{semi}}{6} \|y - x\|_x^3 + \tfrac{\overline{\alpha}}{2} \left\langle \mathbf{B}_x(y - x), (y - x) \right\rangle \tag{29} \\
&= \tfrac{L_{semi}}{6} \|y - x\|_x^3 + \tfrac{\overline{\alpha}}{2} \|y - x\|_{\mathbf{B}_x}^2 \tag{30}
\end{aligned}$$

Representing $\nabla^2 f(x)$-norm in terms of $\mathbf{B}_k$-norm

$$\begin{aligned}
\|y - x\|_x^3 &= \left\langle \nabla^2 f(x)(y - x), y - x \right\rangle^{3/2} \overset{(25)}{\leq} \left\langle (1 + \overline{\alpha})\mathbf{B}_x(y - x), y - x \right\rangle^{3/2} \\
&= (1 + \overline{\alpha})^{3/2} \|y - x\|_{B_x}^3, \tag{31}
\end{aligned}$$

we get for any $x, y \in \mathbb{R}^d$

$$f(y) - \overline{Q}_f(y; x) = \tfrac{L_{semi}}{6} \|y - x\|_x^3 + \tfrac{\overline{\alpha}}{2} \|y - x\|_{\mathbf{B}_x}^2 \overset{(31)}{\leq} \tfrac{(1+\overline{\alpha})^{3/2} L_{semi}}{6} \|y - x\|_{\mathbf{B}_x}^3 + \tfrac{\overline{\alpha}}{2} \|y - x\|_{\mathbf{B}_x}^2.$$

For any $x, y \in \mathbb{R}^d$

$$\begin{aligned}
\overline{Q}_f(y; x) - f(y) &= \overline{Q}_f(y; x) - Q_f(y; x) + Q_f(y; x) - f(y) \\
&\overset{(23)}{\leq} \overline{Q}_f(y; x) - Q_f(y; x) + \tfrac{L_{semi}}{6} \|y - x\|_x^3 \\
&= \tfrac{1}{2} \left\langle (\mathbf{B}_x - \nabla^2 f(x))(y - x), (y - x) \right\rangle + \tfrac{L_{semi}}{6} \|y - x\|_x^3 \\
&\overset{(25)}{\leq} \tfrac{\underline{\alpha}}{2} \left\langle \mathbf{B}_x(y - x), (y - x) \right\rangle + \tfrac{L_{semi}}{6} \|y - x\|_x^3 = \tfrac{\underline{\alpha}}{2} \|y - x\|_{\mathbf{B}_x}^2 + \tfrac{L_{semi}}{6} \|y - x\|_x^3 \\
&\overset{(31)}{\leq} \tfrac{\underline{\alpha}}{2} \|y - x\|_{\mathbf{B}_x}^2 + \tfrac{(1+\overline{\alpha})^{3/2} L_{semi}}{6} \|y - x\|_{\mathbf{B}_x}^3
\end{aligned}$$

For any $x, y \in \mathbb{R}^d$,

$$
\begin{aligned}
\|\nabla f(y) - \nabla \overline{Q}_f(y;x)\|_{\mathbf{B}_x}^* &= \|\nabla f(y) - \nabla Q_f(y;x) + \nabla Q_f(y;x) - \nabla \overline{Q}_f(y;x)\|_{\mathbf{B}_x}^* \\
&= \left\| \nabla f(y) - \nabla f(x) - \nabla^2 f(x)(y-x) + (\nabla^2 f(x) - \mathbf{B}_x)(y-x) \right\|_{\mathbf{B}_x}^* \\
&\leq \left\| \nabla f(y) - \nabla f(x) - \nabla^2 f(x)(y-x) \right\|_{\mathbf{B}_x}^* \\
&\quad + \left\| (\nabla^2 f(x) - \mathbf{B}_x)(y-x) \right\|_{\mathbf{B}_x}^* \\
&\overset{(24)}{\leq} \frac{L_{\text{semi}}}{2}\|y-x\|_x^2 + \left\| (\nabla^2 f(x) - \mathbf{B}_x)(y-x) \right\|_{\mathbf{B}_x}^* \\
&\overset{(25)}{\leq} \frac{(1+\overline{\alpha})^{3/2} L_{\text{semi}}}{2}\|y-x\|_{\mathbf{B}_x}^2 + \left\| (\nabla^2 f(x) - \mathbf{B}_x)(y-x) \right\|_{\mathbf{B}_x}^* \quad (32)
\end{aligned}
$$

For the second term, let $u \overset{\text{def}}{=} \mathbf{B}_x^{1/2}(y-x)$. Then:

$$
\begin{aligned}
\left\| (\nabla^2 f(x) - \mathbf{B}_x)(y-x) \right\|_{\mathbf{B}_x}^* & \\
&= (y-x)^T (\mathbf{B}_x - \nabla^2 f(x)) \mathbf{H}_x (\mathbf{B}_x - \nabla^2 f(x))(y-x) \\
= (y-x)^T \mathbf{B}_x^{1/2} \mathbf{B}_x^{-1/2} (\mathbf{B}_x - \nabla^2 f(x)) \mathbf{B}_x^{-1/2} & \mathbf{B}_x^{-1/2}(\mathbf{B}_x - \nabla^2 f(x)) \mathbf{B}_x^{1/2} \mathbf{B}_x^{-1/2}(y-x) \\
&= u^T (I - \mathbf{B}_x^{-1/2} \nabla^2 f(x) \mathbf{B}_x^{-1/2})^2 u.
\end{aligned}
$$

By (25), we have

$$
-\overline{\alpha} I \preceq (I - \mathbf{B}_x^{-1/2} \nabla^2 f(x) \mathbf{B}_x^{-1/2}) \preceq \underline{\alpha} I \quad \Rightarrow \quad (I - \mathbf{B}_x^{-1/2} \nabla^2 f(x) \mathbf{B}_x^{-1/2})^2 \preceq \alpha_{\max} I.
$$

Therefore,

$$
\left\| (\nabla^2 f(x) - \mathbf{B}_x)(y-x) \right\|_{\mathbf{B}_x}^* \leq \alpha_{\max} u^T u = \alpha_{\max}(y-x)^T \mathbf{B}_x (y-x) = \alpha_{\max}\|y-x\|_{\mathbf{B}_x}.
$$

Plugging this bound into (32) finishes the proof. $\qquad \square$

### B.1 NON-ADAPTIVE METHOD

---
**Algorithm 4** Cubically Enhanced Quasi-Newton Method

---
1: **Requires:** Initial point $x_0 \in \mathbb{R}^d$, constants $L, \theta > 0$.
2: **for** $k = 0, 1, \ldots, K$ **do**
3: $\quad \eta_k = \frac{2}{\theta + \sqrt{\theta^2 + L\|\nabla f(x_k)\|_{\mathbf{H}_k}}}$
4: $\quad x_{k+1} = x_k - \eta_k \mathbf{H}_k \nabla f(x_k)$
5: **Return:** $x_{K+1}$

---

**Theorem 3.** *Let Assumptions 3, 4 hold, $f$ be a convex function, and*

$$
D \overset{def}{=} \max_{k \in [0;K+1]} \|x_k - x_*\|_{\mathbf{B}_k}. \tag{33}
$$

*After $K+1$ iterations of Algorithm 1 with parameters*

$$
\theta \geq 1 + \overline{\alpha}, \; L \geq \frac{(1+\overline{\alpha})^{3/2} L_{semi}}{2}, \tag{34}
$$

*we get the following bound*

$$
f(x_{K+1}) - f(x_*) \leq \frac{(\underline{\alpha}+\overline{\alpha})}{2} \frac{9D^2}{K+3} + (1+\overline{\alpha})^{3/2} \frac{3 L_{semi} D^3}{(K+1)(K+2)}.
$$

*Proof.*

$$f(x_{k+1}) = \min_{y \in \mathbb{R}^d} \left\{ f(x_k) + \tfrac{\theta}{2} \|y - x_k\|_{\mathbf{B}_k}^2 + \tfrac{L}{3} \|y - x_k\|_{\mathbf{B}_k}^3 \right\}$$

$$\overset{(1)}{\leq} \min_{y \in \mathbb{R}^d} \left\{ f(x_k) + \tfrac{1}{2} \|y - x_k\|_{\mathbf{B}_k}^2 + \tfrac{\overline{\alpha}}{2} \|y - x_k\|_{\mathbf{B}_k}^2 + \tfrac{L}{3} \|y - x_k\|_{\mathbf{B}_k}^3 \right\}$$

$$\overset{(4)}{=} \min_{y \in \mathbb{R}^d} \left\{ \overline{Q}_f(y; x_k) + \tfrac{\overline{\alpha}}{2} \|y - x_k\|_{\mathbf{B}_k}^2 + \tfrac{L}{3} \|y - x_k\|_{\mathbf{B}_k}^3 \right\}$$

$$\overset{(11)}{\leq} \min_{y \in \mathbb{R}^d} \left\{ f(y) + \tfrac{\alpha + \overline{\alpha}}{2} \|y - x_k\|_{\mathbf{B}_k}^2 + \tfrac{2L}{3} \|y - x_k\|_{\mathbf{B}_k}^3 \right\}$$

$$\overset{(1)}{\leq} \min_{\gamma_t \in [0,1]} \left\{ f(x_k + \gamma_k(x_* - x_k)) + \tfrac{\alpha + \overline{\alpha}}{2} \gamma_k^2 D^2 + \tfrac{2L}{3} \gamma_k^3 D^3 \right\}$$

$$\overset{\text{convexity}}{\leq} \min_{\gamma_t \in [0,1]} \left\{ (1 - \gamma_k) f(x_k) + \gamma_k f(x_*) + \tfrac{\alpha + \overline{\alpha}}{2} \gamma_k^2 D^2 + \tfrac{2L}{3} \gamma_k^3 D^3 \right\}$$

Subtracting $f(x_*)$ from both sides, we get for any $\gamma_k \in [0, 1]$

$$f(x_{k+1}) - f(x_*) \leq (1 - \gamma_k)(f(x_k) - f(x_*)) + \tfrac{\alpha + \overline{\alpha}}{2} \gamma_k^2 D^2 + \tfrac{2L}{3} \gamma_k^3 D^3. \tag{35}$$

Let us select $\gamma_0 = 1$ and define sequence $A_k$

$$A_k \overset{\text{def}}{=} \begin{cases} 1, & k = 0 \\ \prod_{i=1}^{k} (1 - \gamma_i), & k \geq 1. \end{cases}$$

Then $A_k = (1 - \eta_k) A_{t-k}$. Dividing both sides of (35) by $A_k$, we get

$$\tfrac{1}{A_k}(f(x_{k+1}) - f(x_*)) \leq \tfrac{(1 - \gamma_k)}{A_k}(f(x_k) - f(x_*)) + \tfrac{\alpha + \overline{\alpha}}{2} \tfrac{\gamma_k^2}{A_k} D^2 + \tfrac{2L}{3} \tfrac{\gamma_k^3}{A_k} D^3$$

$$= \tfrac{1}{A_{k-1}}(f(x_k) - f(x_*)) + \tfrac{\alpha + \overline{\alpha}}{2} \tfrac{\gamma_k^2}{A_k} D^2 + \tfrac{2L}{3} \tfrac{\gamma_k^3}{A_k} D^3.$$

Summing both sides of inequality above from $k = 0, \ldots, K$, we obtain

$$\tfrac{1}{A_K}(f(x_{K+1}) - f(x_*)) \leq \tfrac{1 - \gamma_0}{A_0}(f(x_0) - f(x_*)) + \tfrac{\alpha + \overline{\alpha}}{2} D^2 \sum_{k=0}^{K} \tfrac{\gamma_k^2}{A_k} + \tfrac{2L}{3} D^3 \sum_{k=0}^{K} \tfrac{\gamma_k^3}{A_k}. \tag{36}$$

Let us choose $\gamma_k = \tfrac{3}{k+3}$. By [(2.23), Ghadimi et al. (2017)], we have

$$A_k = \frac{6}{(k+1)(k+2)(k+3)}, \quad \sum_{k=0}^{K} \frac{\gamma_k^2}{A_k} \leq \frac{3(K+1)(K+2)}{2}, \quad \sum_{k=0}^{K} \frac{\gamma_k^3}{A_k} \leq \frac{3K}{2}. \tag{37}$$

$$f(x_{K+1}) - f(x_*) \overset{(36), \gamma_0 = 1}{=} A_K \frac{(\alpha + \overline{\alpha}) D^2}{2} \frac{3(K+1)(K+2)}{2} + A_K \frac{2L D^3}{3} \frac{3K}{2}$$

$$\overset{(37)}{=} \frac{(\alpha + \overline{\alpha})}{2} \frac{9 D^2}{K+3} + \frac{6 L D^3}{(K+1)(K+2)} \overset{(1)}{\leq} \frac{(\alpha + \overline{\alpha})}{2} \frac{9 D^2}{K+3} + (1 + \overline{\alpha})^{3/2} \frac{3 L_{\text{semi}} D^3}{(K+1)(K+2)}$$

$$\square$$

**Lemma 7.** *Let Assumptions 3, 4 hold and $f(x)$ be a convex function. Quasi-Newton methods with CEQN stepsize with parameters $\theta = 1 + \alpha \geq 1 + \alpha_{max}$, $L \geq (1 + \overline{\alpha})^{3/2} L_{semi}$ implies the following one-step decrease*

$$f(x_k) - f(x_{k+1}) \geq \min \left\{ \tfrac{1}{4\alpha} \|\nabla f(x_{k+1})\|_{\mathbf{B}_k}^{*2}, \left( \tfrac{1}{6L} \right)^{\frac{1}{2}} \|\nabla f(x_{k+1})\|_{\mathbf{B}_k}^{*\frac{3}{2}} \right\} \geq 0. \tag{38}$$

*Proof.* By optimality condition of CEQN regularized model

$$0 = \nabla \overline{Q}(x_{k+1}, x_k) + (\theta - 1 + L \|x_{k+1} - x_k\|_{\mathbf{B}_k}) \mathbf{B}_k (x_{k+1 - x_k}). \tag{39}$$

Let us define $\zeta_k \overset{\text{def}}{=} \alpha + L\|x_{k+1} - x_k\|_{\mathbf{B}_k}$, $\overline{L} = \frac{L_{\text{semi}}}{2}(1+\overline{\alpha})^{3/2}$, where $\alpha = \theta - 1$. Next,

$$
(\alpha_{\max} + \overline{L}\|x_{k+1} - x_k\|_{\mathbf{B}_k})^2\|x_{k+1} - x_k\|_{\mathbf{B}_k}^2
$$
$$
\overset{(12)}{\geq} \|\nabla \overline{Q}_f(x_{k_1}; x_k) - f(x_{k+1})\|_{\mathbf{B}_k}^{*2} \overset{(39)}{=} \|\zeta_k \mathbf{B}_k(x_{k+1} - x_k) + \nabla f(x_{k+1})\|_{\mathbf{B}_k}^{*2}
$$
$$
= \zeta_k^2\|x_{k+1} - x_k\|_{\mathbf{B}_k}^2 + \|\nabla f(x_{k+1})\|_{\mathbf{B}_k}^{*2} + 2\zeta_k\langle\nabla f(x_{k+1}), x_{k+1} - x_k\rangle. \tag{40}
$$

We consider two cases, based on which term in $\zeta_k$ dominates.

- Let $\alpha \geq L\|x_{k+1} - x_k\|_{\mathbf{B}_k}$. Then $\zeta_k \leq 2\alpha$. By the choice of the parameters, we have

$$
\zeta_k^2\|x_{k+1} - x_k\|_{\mathbf{B}_k} \geq \left(\alpha_{\max} + \frac{(1+\overline{\alpha})^{3/2}L_{\text{semi}}}{2}\|x_{k+1} - x_k\|_{\mathbf{B}_k}\right)^2\|x_{k+1} - x_k\|_{\mathbf{B}_k}^2
$$
$$
\overset{(40)}{\geq} \zeta_k^2\|x_{k+1} - x_k\|_{\mathbf{B}_k}^2 + \|\nabla f(x_{k+1})\|_{\mathbf{B}_k}^{*2} + 2\zeta_k\langle\nabla f(x_{k+1}), x_{k+1} - x_k\rangle.
$$

Therefore,

$$
\langle\nabla f(x_{k+1}), x_k - x_{k+1}\rangle \geq \frac{1}{2\zeta_k}\|\nabla f(x_{k+1})\|_{\mathbf{B}_k}^{*2} \geq \frac{1}{4\alpha}\|\nabla f(x_{k+1})\|_{\mathbf{B}_k}^{*2}. \tag{41}
$$

- Now, let $\alpha < L\|x_{k+1} - x_k\|_{\mathbf{B}_k}$. Then $\zeta_k < 2L\|x_{k+1} - x_k\|_{\mathbf{B}_k}$.
  From

$$
(\alpha_{\max} + \overline{L}\|x_{k+1} - x_k\|_{\mathbf{B}_k})^2\|x_{k+1} - x_k\|_{\mathbf{B}_k}^2
$$
$$
\overset{(40)}{\geq} \zeta_k^2\|x_{k+1} - x_k\|_{\mathbf{B}_k}^2 + \|\nabla f(x_{k+1})\|_{\mathbf{B}_k}^{*2} + 2\zeta_k\langle\nabla f(x_{k+1}), x_{k+1} - x_k\rangle
$$

and our choice of parameters, we get

$$
\langle\nabla f(x_{k+1}), x_k - x_{k+1}\rangle
$$
$$
\geq \frac{\|\nabla f(x_{k+1})\|_{\mathbf{B}_k}^{*2}}{2\zeta_k} + \left[\zeta_k^2 - (\alpha_{\max} + \overline{L}\|x_{k+1} - x_k\|_{\mathbf{B}_k}^2)\right]\frac{\|x_{k+1} - x_k\|_{\mathbf{B}_k}^2}{2\zeta_k}
$$
$$
= \frac{\|\nabla f(x_{k+1})\|_{\mathbf{B}_k}^{*2}}{2\zeta_k}
$$
$$
+ (\alpha - \alpha_{\max} + \frac{L-\overline{L}}{2}\|x_{k+1} - x_k\|_{\mathbf{B}_k})(\alpha + \alpha_{\max} + \frac{L+\overline{L}}{2}\|x_{k+1} - x_k\|_{\mathbf{B}_k})\frac{\|x_{k+1} - x_k\|_{\mathbf{B}_k}^2}{2\zeta_k}
$$
$$
\geq \frac{\|\nabla f(x_{k+1})\|_{\mathbf{B}_k}^{*2}}{4L\|x_{k+1} - x_k\|_{\mathbf{B}_k}} + \frac{L^2 - \overline{L}^2}{4L}\|x_{k+1} - x_k\|_{\mathbf{B}_k}^3
$$
$$
= \frac{\|\nabla f(x_{k+1})\|_{\mathbf{B}_k}^{*2}}{4L\|x_{k+1} - x_k\|_{\mathbf{B}_k}} + \frac{3L}{16}\|x_{k+1} - x_k\|_{\mathbf{B}_k}^3
$$
$$
\geq \left(\frac{1}{6L}\right)^{\frac{1}{2}}\|\nabla f(x_{k+1})\|_{\mathbf{B}_k}^{*\frac{3}{2}}, \tag{42}
$$

where for the last inequality, we use $\frac{\alpha}{r} + \frac{\beta r^3}{3} \geq \frac{4}{3}\beta^{1/4}\alpha^{3/4}$.

By combing results of these cases and using convexity $f(x_k) - f(x_{k+1}) \geq \langle\nabla f(x_{k+1}), x_k + x_{k+1}\rangle$ we get desired bound. $\qquad\square$

**Remark 3.** *Let Assumptions 3, 4 hold, and*

$$
D \overset{def}{=} \max_{k\in[0;K+1]}\|x_k - x_*\|_{\mathbf{B}_k},
$$

*where $x_k$ are the iterates generated by Algorithm 1, and let $\underline{\alpha} < 1$. Assume that the level set of $f$ is bounded:*

$$
\max_{x\in\mathcal{L}(x_0)}\|x - x_*\| \leq R < \infty,
$$

where $\mathcal{L}(x_0) = \{x \mid f(x) \leq f(x_0)\}$. *Then*

$$D \leq (1 - \underline{\alpha})^{-1/2} \max_{x \in \mathcal{L}(x_0)} \left\{ \left( \||x - x_*||_{x_*}^2 + L_{semi} \|x - x_*\|_{x_*}^3 \right)^{1/2} \right\}$$

$$\leq (1 - \underline{\alpha})^{-1/2} (R^2 \|\nabla^2 f(x_*)\| + L_{semi} R^3 \|\nabla^2 f(x_*)\|^{3/2})^{1/2}.$$

*Proof.* By Assumption 4 $(1 - \alpha)B_k \preceq \nabla^2 f(x_k)$. Thus,

$$\|x_k - x_*\|_{B_k} \leq (1 - \underline{\alpha})^{-1/2} \|x_k - x_*\|_k.$$

Next, we bound $\|x_k - x_*\|_k$. By Lemma 7 we have $f(x_0) \geq f(x_1) \geq ... \geq f(x_k) \geq f(x_{k+1}) \geq ... \geq f(x_{K+1})$, hence $\{x_i\}_{i=0}^{K+1} \subseteq \mathcal{L}(x_0)$. By Assumption 3,

$$(x_k - x_*)^T (\nabla^2 f(x_k) - \nabla^2 f(x_*))(x_k - x_*) \leq L_{\text{semi}} \|x_k - x_*\|_{x_*}^3.$$

Therefore,

$$\|x_k - x_*\|_k^2 \leq \|x_k - x_*\|_{x_*}^2 + L_{\text{semi}} \|x_k - x_*\|_{x_*}^3$$

$$\leq \max_{x \in \mathcal{L}(x_0)} \left\{ \|x - x_*\|_{x_*}^2 + L_{\text{semi}} \|x - x_*\|_{x_*}^3 \right\},$$

$$\leq R^2 \|\nabla^2 f(x_*)\| + L_{\text{semi}} R^3 \|\nabla^2 f(x_*)\|^{3/2}.$$

which depends only on $R, \|\nabla^2 f(x_*)\|, L_{\text{semi}}$. $\qquad\qquad\square$

**Corollary 3.** *Let Assumptions 3, 4 hold and $f$ be a convex function. Algorithm 1 with parameters $\theta = 1 + \alpha \geq 1 + \alpha_{max}$, $L \geq (1 + \overline{\alpha})^{3/2} L_{semi}$ converges with the rate*

$$f(x_{k+1}) - f(x^*) \leq \frac{270(1 + \alpha)^{3/2} L_{semi} \overline{D}^3}{k^2}.$$

*until it reaches the region $\|\nabla f(x_{k+1})\|_{\mathbf{B}_k}^* \leq \frac{4\alpha^2}{9L^2(1+\alpha)^{3/2}}$, where*

$$\overline{D} \overset{def}{=} \max_{k \in [0; K+1]} \left( \|x_k - x_*\|_{\mathbf{B}_k} + \|\nabla f(x_k)\|_{\mathbf{B}_k}^* \right). \tag{43}$$

*Proof.* Let us assume that $\frac{1}{4\alpha} \|\nabla f(x_{k+1})\|_{\mathbf{B}_k}^{*2} \geq \left(\frac{1}{6L}\right)^{\frac{1}{2}} \|\nabla f(x_{k+1})\|_{\mathbf{B}_k}^{*\frac{3}{2}}$. Then, $\|\nabla f(x_{k+1})\|_{\mathbf{B}_k}^* \geq \frac{8}{3} \frac{\alpha^2}{L} \geq \frac{8}{3} \frac{\alpha^2}{(1+\alpha)^{3/2} L_{\text{semi}}}$. Then, by Lemma 2

$$f(x_k) - f(x_{k+1}) \geq \frac{1}{\sqrt{6L}} \|\nabla f(x_{k+1})\|_{\mathbf{B}_k}^{*\frac{3}{2}}$$

By convexity, we get

$$f(x^*) \geq f(x_{t+1}) + \langle \nabla f(x_{t+1}), x^* - x_{t+1} \rangle \geq f(x_{t+1}) - \|\nabla f(x_{k+1})\|_{\mathbf{B}_k}^* \|x^* - x_{k+1}\|_{\mathbf{B}_k}.$$

Hence,

$$\|\nabla f(x_{k+1})\|_{\mathbf{B}_k}^* \geq \frac{f(x_{t+1}) - f(x^*)}{\|x^* - x_{k+1}\|_{\mathbf{B}_k}}. \tag{44}$$

By the definition of CEQN step, $\eta_k \leq 1$, and (15)

$$\|x^* - x_{k+1}\|_{\mathbf{B}_k} = \left\|x^* - x_k + \eta_k \mathbf{B}_k^{-1} \nabla f(x_k)\right\|_{\mathbf{B}_k} \leq \|x^* - x_k\|_{\mathbf{B}_k} + \|\nabla f(x_k)\|_{\mathbf{B}_k}^* \leq \overline{D}.$$

Then,

$$f(x_k) - f(x_{k+1}) \geq \left(\frac{f(x_{k+1}) - f(x^*)}{\overline{D}}\right)^{3/2} \left(\frac{1}{6L}\right)^{1/2} \geq \left(\frac{f(x_{k+1}) - f(x^*)}{\overline{D}}\right)^{3/2} \left(\frac{1}{6L_{\text{semi}}(1+\alpha)^{3/2}}\right)^{1/2}.$$

By setting $\xi_k = \frac{f(x_k) - f(x^*)}{6L_{\text{semi}}(1+\alpha)^{3/2}}$ we get the following condition

$$\xi_k - \xi_{k+1} \geq \xi_{k+1}^{3/2}.$$

In Nesterov (2022)[Lemma A.1] it is shown that if a non–negative sequence $\{\xi_t\}$ satisfies for $\beta > 0$

$$\xi_k - \xi_{k+1} \geq \xi_{k+1}^{1+\beta}$$

then for all $k \geq 0$

$$\xi_k \leq \left[\left(1 + \frac{1}{\beta}\right)\left(1 + \xi_0^\beta\right)\frac{1}{k}\right]^{1/\beta}. \tag{45}$$

Then, by (45) we get

$$\xi_{k+1} \leq \left[3\left(1 + \frac{f(x_1) - f(x^*)}{6L_{\text{semi}}\overline{D}^3(1+\alpha)^{3/2}}\right)\frac{1}{k}\right]^2.$$

Therefore,

$$f(x_{k+1}) - f(x^*) \leq \frac{54(1+\alpha)^{3/2}L_{\text{semi}}\overline{D}^3}{k^2} + \frac{9}{k^2}(f(x_1) - f(x^*)). \tag{46}$$

Now, we consider the second term

$$\left(\frac{1}{6L_{\text{semi}}(1+\alpha)^{3/2}}\right)^{1/2}\|\nabla f(x_1)\|_{\mathbf{B}_0}^{*}{}^{3/2} \stackrel{(44)}{\leq} \langle \nabla f(x_1), x_0 - x_1\rangle \leq \|\nabla f(x_1)\|_{\mathbf{B}_0}^{*}\|x_0 - x_1\|_{\mathbf{B}_0}$$

$$\leq \|\nabla f(x_1)\|_{\mathbf{B}_0}^{*}\left(\|x_0 - x^*\|_{\mathbf{B}_0} + \left\|x_0 - \eta_k\mathbf{B}_0^{-1}\nabla f(x_0) - x^*\right\|_{\mathbf{B}_0}\right)$$

$$\leq \|\nabla f(x_1)\|_{\mathbf{B}_0}^{*}\left(\|x_0 - x^*\|_{\mathbf{B}_0} + \|x_0 - x^*\|_{\mathbf{B}_0} + \|\nabla f(x_0)\|_{\mathbf{B}_0}^{*}\right)$$

$$\leq 2\|\nabla f(x_1)\|_{\mathbf{B}_0}^{*}\overline{D}. \tag{47}$$

Next, by convexity, we get

$$f(x_1) - f(x^*) \leq \overline{D}\|\nabla f(x_1)\|_{\mathbf{B}_0}^{*} \stackrel{(47)}{\leq} 24(1+\alpha)^{3/2}L_{\text{semi}}\overline{D}^3. \tag{48}$$

And by using (46), we obtain convergence rate

$$f(x_{k+1}) - f(x^*) \leq \frac{54(1+\alpha)^{3/2}L_{\text{semi}}\overline{D}^3}{k^2} + \frac{9}{k^2}(f(x_1) - f(x^*)) \leq \frac{270(1+\alpha)^{3/2}L_{\text{semi}}\overline{D}^3}{k^2}.$$

$\square$

**Remark 4.** *Let Assumptions 3, 4 hold and let $\underline{\alpha} < 1$. Assume that the level set of $f$ is bounded (14). Then $\overline{D}$ depends only on the constants $\underline{\alpha}$, $R$, $L_{semi}$, $\|\nabla^2 f(x_*)\|$.*

*Proof.* By the definition

$$\overline{D} \stackrel{\text{def}}{=} \max_{k \in [0;K+1]}\left(\|x_k - x_*\|_{\mathbf{B}_k} + \|\nabla f(x_k)\|_{\mathbf{B}_k}^{*}\right) \leq \max_{k \in [0;K+1]}\|x_k - x_*\|_{\mathbf{B}_k} + \max_{k \in [0;K+1]}\|\nabla f(x_k)\|_{\mathbf{B}_k}^{*}.$$

In the proof of Remark 1 the first term was bounded and shown that it depends only on the constants $\underline{\alpha}$, $R$, $L_{\text{semi}}$, $\|\nabla^2 f(x_*)\|$.

Then, lets focus on the gradient term. By triangle inequality and Assumption 3

$$\|\nabla f(x_k)\|_k^{*} \leq \|\nabla f(x_k) + \nabla^2 f(x_k)(x^* - x_k)\|_k^{*} + \|\nabla^2 f(x_k)(x^* - x_k)\|_k^{*}$$

$$\leq \frac{L_{\text{semi}}}{2}\|x^* - x_k\|_k^2 + \|x^* - x_k\|_k.$$

The term $\|x^* - x_k\|_k$ was bounded in the proof of Remark 1. Finally, by Assumption 4

$$\|\nabla f(x_k)\|_{\mathbf{B}_k}^{*} \leq (1 + \overline{\alpha})^{1/2}\|\nabla f(x_k)\|_k^{*}.$$

Which proves this remark. $\square$

**Corollary 4.** *Let Assumptions 3 and 4 hold, and let $f$ be a convex function. Suppose Algorithm 1 is run with parameters $\theta_k = 1 + \alpha_k \geq 1 + \alpha_{\max}$ and $L \geq (1 + \overline{\alpha})^{3/2}L_{semi}$. If the inexactness satisfies $\alpha_k \leq L\|x_{k+1} - x_k\|_{\mathbf{B}_k}$, then Algorithm 1 achieves the convergence rate*

$$f(x_{k+1}) - f(x^*) \leq \frac{270(1+\alpha)^{3/2}L_{semi}\overline{D}^3}{k^2}.$$

*Proof.* If $\alpha_k \leq L\|x_{k+1} - x_k\|_{\mathbf{B}_k}$, then following the proof of Lemma 2, we arrive at

$$f(x_k) - f(x_{k+1}) \geq \frac{1}{\sqrt{6L}} \|\nabla f(x_{k+1})\|_{\mathbf{B}_k}^{*\frac{3}{2}}.$$

Then, directly by the proof of Lemma 1 we achieve the desired bound with $\alpha = \max\limits_k \alpha_k$.

$\square$

### B.2 ADAPTIVE METHOD

---

**Algorithm 5** Adaptive Cubically Enhanced Quasi-Newton Method

---

1: **Requires:** Initial point $x_0 \in \mathbb{R}^d$, constant $L$ s.t. $L \geq 2L > 0$, initial inexactness $\alpha_0 > 0$, increase multiplier $\gamma_{inc} > 1$.
2: **for** $k = 0, 1, \ldots, K$ **do**
3:    Calculate stepsize

$$\eta_k = \frac{2}{(1+\alpha_k) + \sqrt{(1+\alpha_k)^2 + (1+\alpha_k)^{3/2} L \|\nabla f(x_k)\|_{\mathbf{H}_k}^*}} \tag{49}$$

4:    Perform Quasi-Newton step

$$x_{k+1} = x_k - \eta_k \mathbf{H}_k \nabla f(x_k) \tag{50}$$

5:    **while** $\langle \nabla f(x_{t+1}), x_k - x_{k+1} \rangle \leq \min \left\{ \frac{\|\nabla f(x_{k+1})\|_{\mathbf{B}_k}^{*2}}{4\alpha_k}, \frac{\|\nabla f(x_{t+1})\|_{\mathbf{B}_k}^{*\frac{3}{2}}}{(6(1+\alpha_k)^{3/2} L)^{1/2}} \right\}$ **do**
6:        $\alpha_k = \alpha_k \gamma_{inc}$
7:        Calculates stepsize $\eta_k$ (49) with updated $\alpha_k$
8:        Perform Quasi-Newton step (50) with updated $\eta_k$
9: **Return:** $x_{T+1}$

---

**Theorem 4.** *Let Assumptions 3, 4. After $K + 1$ iterations of Algorithm* (2) *with parameters $L \geq 2L_{semi}$, $\alpha_0 > 0$, $\gamma_{inc} > 0$. Let $\varepsilon > 0$ be the desired solution accuracy. Then after*

$$K = O\left(\frac{\alpha_K \overline{D}^2}{\varepsilon} + \frac{(1 + \alpha_K)^{3/2} L \overline{D}^3}{\sqrt{\varepsilon}} + \log_{\gamma_{inc}}\left(\frac{\alpha_{\max}}{\alpha_0}\right)\right)$$

*iterations of Algorithm 2 $x_K$ is an $\varepsilon$-solution, i.e. $f(x_K) - f(x^*) \leq \varepsilon$.*

*Proof.* By Lemma 2, the termination condition on Line 4 of Algorithm 2 is guaranteed to be satisfied after a finite number of backtracking steps. Specifically, the number of inner iterations is bounded by $\log_{\gamma_{inc}}\left(\frac{\alpha_{\max}}{\alpha_0}\right)$. Denoting $L_k = (1 + \alpha_k)^{3/2} L$, we obtain the following bound for each iteration:

$$\langle \nabla f(x_{k+1}), x_k - x_{k+1} \rangle \geq \min \left\{ \left(\frac{1}{4\alpha_k}\right) \|\nabla f(x_{k+1})\|_{\mathbf{B}_k}^{*2}, \left(\frac{1}{6L_k}\right)^{\frac{1}{2}} \|\nabla f(x_{k+1})\|_{\mathbf{B}_k}^{*\frac{3}{2}} \right\}. \tag{51}$$

Since $f(x)$ is convex

$$\begin{aligned}
f(x_t) - f(x_{t+1}) &\geq \langle \nabla f(x_{t+1}), x_t - x_{t+1} \rangle \\
&\overset{(51)}{\geq} \min \left\{ \left(\frac{1}{4\alpha_k}\right) \|\nabla f(x_{k+1})\|_{\mathbf{B}_k}^{*2}, \left(\frac{1}{6L_k}\right)^{\frac{1}{2}} \|\nabla f(x_{k+1})\|_{\mathbf{B}_k}^{*\frac{3}{2}} \right\} \geq 0. \tag{52}
\end{aligned}$$

Thus, the method produces a monotonically non-increasing sequence of function values, with strict decrease whenever $\nabla f(x_{k+1}) \neq 0$. Once $\nabla f(x_k) = 0$, we have $x_{k+1} = x^*$ and the method converged. Furthermore, by convexity, we get

$$f(x^*) \geq f(x_{t+1}) + \langle \nabla f(x_{t+1}), x^* - x_{t+1} \rangle \geq f(x_{t+1}) - \|\nabla f(x_{k+1})\|_{\mathbf{B}_k}^* \|x^* - x_{k+1}\|_{\mathbf{B}_k}.$$

Hence,

$$\|\nabla f(x_{k+1})\|_{\mathbf{B}_k}^* \geq \frac{f(x_{t+1}) - f(x^*)}{\|x^* - x_{k+1}\|_{\mathbf{B}_k}}. \tag{53}$$

By the definition of CEQN step, $\eta_k \leq 1$, and (15)

$$\|x^* - x_{k+1}\|_{\mathbf{B}_k} = \left\|x^* - x_k + \eta_k \mathbf{B}_k^{-1} \nabla f(x_k)\right\|_{\mathbf{B}_k} \leq \|x^* - x_k\|_{\mathbf{B}_k} + \|\nabla f(x_k)\|_{\mathbf{B}_k}^* \leq \overline{D}.$$

Then, we get $\|x^* - x_{k+1}\|_{\mathbf{B}_k} \geq \frac{f(x_{t+1}) - f(x^*)}{\overline{D}}$. Therefore, by combining with (52), we have

$$f(x_k) - f(x_{k+1}) \geq \min\left\{ \left(\frac{f(x_{k+1}) - f(x^*)}{\overline{D}}\right)^2 \left(\frac{1}{4\alpha_k}\right), \left(\frac{f(x_{k+1}) - f(x^*)}{\overline{D}}\right)^{\frac{3}{2}} \left(\frac{1}{6L_k}\right)^{\frac{1}{2}} \right\}. \tag{54}$$

Next, we aim to show that Quasi-Newton methods with the Adaptive CEQN stepsize exhibit two convergence regimes, with at most one switch between them.

We begin by analyzing the case when the minimum on the right-hand side of (54) is attained by the second term. This occurs when

$$f(x_{k+1}) - f(x^*) \geq \frac{8}{3} \frac{\alpha_k^2 \overline{D}}{L_k} = \frac{8}{3} \frac{\alpha_k^2 \overline{D}}{(1 + \alpha_k)^{3/2} L}.$$

Note that $\alpha_k$ is monotonically increasing by the design of the algorithm. Therefore, the right-hand side of the inequality is also increasing in $k$, while the left-hand side, $f(x_{t+1}) - f(x^*)$, is monotonically decreasing. Consequently, the inequality can be violated at most once, implying that the switch between regimes can occur only once. Let us denote number of iteration in the first regime as $K_1 \geq 0$, $K_2 \geq 0$ in the second regime, and $K = K_1 + K_2$ total number of iterations of outer step of the method. Total number of Quasi-Newton method with CEQN stepsize would be $K + \log_{\gamma_{\text{inc}}}\left(\frac{\alpha_{\max}}{\alpha_0}\right)$.

At first, Algorithm 2 performs $K_1 \geq 0$ iterations with the following guarantee:

$$f(x_k) - f(x_{k+1}) \geq \left(\frac{f(x_{k+1}) - f(x^*)}{\overline{D}}\right)^{3/2} \left(\frac{1}{6L_k}\right)^{1/2} \geq \left(\frac{f(x_{k+1}) - f(x^*)}{\overline{D}}\right)^{3/2} \left(\frac{1}{6L(1+\alpha_K)^{3/2}}\right)^{1/2}.$$

Then, we have

$$f(x_k) - f(x_{k+1}) \geq \left(\frac{f(x_{k+1}) - f(x^*)}{\overline{D}}\right)^{3/2} \left(\frac{1}{6L_k}\right)^{1/2} \geq \left(\frac{f(x_{k+1}) - f(x^*)}{\overline{D}}\right)^{3/2} \left(\frac{1}{6L(1+\alpha_K)^{3/2}}\right)^{1/2}.$$

By setting $\xi_k = \frac{f(x_k) - f(x^*)}{6L(1+\alpha_K)^{3/2}}$ we get the following condition

$$\xi_k - \xi_{k+1} \geq \xi_{k+1}^{3/2}.$$

In Nesterov (2022)[Lemma A.1] it is shown that if a non–negative sequence $\{\xi_t\}$ satisfies for $\beta > 0$

$$\xi_k - \xi_{k+1} \geq \xi_{k+1}^{1+\beta}$$

then for all $k \geq 0$

$$\xi_k \leq \left[\left(1 + \frac{1}{\beta}\right)\left(1 + \xi_0^\beta\right)\frac{1}{k}\right]^{1/\beta}. \tag{55}$$

Then, by (55) we get for $k \in [0, K_1]$

$$\xi_{k+1} \leq \left[3\left(1 + \frac{f(x_1) - f(x^*)}{6L\overline{D}^3(1+\alpha_k)^{3/2}}\right)\frac{1}{k}\right]^2.$$

Therefore,

$$f(x_{k+1}) - f(x^*) \leq \frac{54(1 + \alpha_K)^{3/2} L\overline{D}^3}{k^2} + \frac{9}{k^2}(f(x_1) - f(x^*)). \tag{56}$$

$$\left(\frac{1}{6L(1+\alpha_K)^{3/2}}\right)^{1/2} \|\nabla f(x_1)\|_{\mathbf{B}_0}^{*\frac{3}{2}} \overset{(53)}{\leq} \langle \nabla f(x_1), x_0 - x_1 \rangle \leq \|\nabla f(x_1)\|_{\mathbf{B}_0}^* \|x_0 - x_1\|_{\mathbf{B}_0}$$
$$\leq \|\nabla f(x_1)\|_{\mathbf{B}_0}^* (\|x_0 - x^*\|_{\mathbf{B}_0} + \|x_0 - \eta_k \mathbf{B}_0^{-1}\nabla f(x_0) - x^*\|_{\mathbf{B}_0})$$
$$\leq \|\nabla f(x_1)\|_{\mathbf{B}_0}^* (\|x_0 - x^*\|_{\mathbf{B}_0} + \|x_0 - x^*\|_{\mathbf{B}_0} + \|\nabla f(x_0)\|_{\mathbf{B}_0}^*)$$
$$\leq 2\|\nabla f(x_1)\|_{\mathbf{B}_0}^* \overline{D}. \tag{57}$$

Next, by convexity, we get

$$f(x_1) - f(x^*) \le \overline{D}\|\nabla f(x_1)\|^*_{\mathbf{B}_0} \overset{(57)}{\le} 24(1+\alpha_K)^{3/2}L\overline{D}^3. \tag{58}$$

And by using (56), we obtain convergence rate

$$f(x_{k+1}) - f(x^*) \le \frac{54(1+\alpha_K)^{3/2}L\overline{D}^3}{k^2} + \frac{9}{k^2}(f(x_1) - f(x^*)) \le \frac{270(1+\alpha_K)^{3/2}L\overline{D}^3}{k^2}.$$

Equivalently,

$$K_1 = O\left(\frac{(1+\alpha_K)^{3/2}L\overline{D}^3}{\sqrt{\varepsilon}}\right).$$

After $K_1$ iterations, if the target accuracy has not yet been achieved, the method transitions into the second regime. From (54), similar to the first regime, for $k \in [K_1, K]$, we get

$$\xi_k - \xi_{k+1} \ge \xi_{k+1}^2, \tag{59}$$

with $\xi_k = \frac{f(x_k) - f(x^*)}{4\alpha_K \overline{D}^2}$.

Applying (55), we get

$$f(x_K) - f(x^*) \le \left(4\alpha_K\overline{D}^2 + f(x_{K_1}) - f(x^*)\right)\frac{2}{K_2} \tag{60}$$

$$\left(\tfrac{1}{4\alpha_K}\right)^{1/2}\|\nabla f(x_{K_1})\|^{*~3/2}_{\mathbf{B}_{K_1-1}} \tag{61}$$

$$\overset{(53)}{\le} \langle\nabla f(x_{K_1}), x_{K_1-1} - x_{K_1}\rangle \le \|\nabla f(x_{K_1})\|^*_{\mathbf{B}_{K_1-1}}\|x_{K_1-1} - x_{K_1}\|_{\mathbf{B}_{K_1-1}}$$

$$\le \|\nabla f(x_1)\|^*_{\mathbf{B}_{K_1-1}}\left(\|x_{K_1-1} - x^*\|_{\mathbf{B}_{K_1-1}} + \|x_{K_1-1} - \eta_k\mathbf{B}_{K_1-1}^{-1}\nabla f(x_{K_1-1}) - x^*\|_{\mathbf{B}_{K_1-1}}\right)$$

$$\le \|\nabla f(x_{K_1})\|^*_{\mathbf{B}_{K_1-1}}\left(\|x_{K_1-1} - x^*\|_{\mathbf{B}_{K_1-1}} + \|x_{K_1-1} - x^*\|_{\mathbf{B}_{K_1-1}} + \|\nabla f(x_{K_1-1})\|^*_{\mathbf{B}_{K_1-1}}\right)$$

$$\le 2\|\nabla f(x_{K_1})\|^*_{\mathbf{B}_{K_1-1}}\overline{D}. \tag{62}$$

Next, by convexity, we get

$$f(x_{K_1}) - f(x^*) \le \overline{D}\|\nabla f(x_{K_1})\|^*_{\mathbf{B}_{K_1-1}} \overset{(57)}{\le} 8\alpha_K\overline{D}^2. \tag{63}$$

And by using (60), we obtain convergence rate

$$f(x_K) - f(x^*) \le \frac{24\alpha_K\overline{D}^2}{k}$$

Equivalently,

$$K_2 = O\left(\frac{\alpha_K\overline{D}^2}{\varepsilon}\right).$$

Thus, total number of iterations is

$$K_1 + K_2 + \log_{\gamma_{\text{inc}}}\left(\frac{\alpha_{\max}}{\alpha_0}\right) = O\left(\frac{\alpha_K\overline{D}^2}{\varepsilon} + \frac{(1+\alpha_K)^{3/2}L\overline{D}^3}{\sqrt{\varepsilon}} + \log_{\gamma_{\text{inc}}}\left(\frac{\alpha_{\max}}{\alpha_0}\right)\right)$$

$$\square$$

**Lemma 8.** *Let Assumptions 3, 4 hold. QN step with CEQN stepsize and with parameters $\theta \ge 1 + \alpha_{max}$, $L \ge (1+\overline{\alpha})^{3/2}L_{semi}$ implies one-step decrease*

$$f(x_{k+1}) \le f(x_k) - \tfrac{1}{2}\eta_k\left(\|\nabla f(x_k)\|^*_{\mathbf{B}_k}\right)^2 - \tfrac{L}{6}\eta_k^3\left(\|\nabla f(x_k)\|^*_{\mathbf{B}_k}\right)^3 \tag{64}$$

*Proof.* By Lemma 4 for any $x, \, y \in \mathbb{R}^d$

$$|f(y) - f(x) - \langle \nabla f(x), y - x \rangle - \tfrac{1}{2} \langle \nabla^2 f(x)(y - x), y - x \rangle| \leq \tfrac{L_{\text{semi}}}{6} \|y - x\|_x^3 \qquad (65)$$

Substituting $y = x_k, \, x = x_{k+1}$:

$$f(x_k) - f(x_{k+1}) \geq \langle \nabla f(x_k), x_k - x_{k+1} \rangle - \tfrac{1}{2} \langle \nabla^2 f(x_k)(x_{k+1} - x_k), x_{k+1} - x_k \rangle - \frac{L_{\text{semi}}}{6} \|x_{k+1} - x_k\|_{x_k}^3. \qquad (66)$$

From optimality condition of the cubic step

$$0 = \nabla f(x_k) + \theta B_k(x_{k+1} - x_k) + \frac{2L}{3} \|x_{k+1} - x_k\|_{\mathbf{B}_k} \mathbf{B}_k(x_{k+1} - x_k). $$

Multiplying optimality condition by $\frac{1}{2}(x_{k+1} - x_k)^\top$:

$$0 = \frac{1}{2} \langle \nabla f(x_k), x_{k+1} - x_k \rangle + \frac{\theta}{2} \|x_{k+1} - x_k\|_{\mathbf{B}_k}^2 + \frac{L}{3} \|x_{k+1} - x_k\|_{\mathbf{B}_k}^3 \qquad (67)$$

By Assumption 4 and our parameters choice, we have

$$-\frac{1}{2} \|x_{k+1} - x_k\|_{x_k}^2 \geq -\frac{1 + \overline{\alpha}}{2} \|x_{k+1} - x_k\|_{\mathbf{B}_k}^2 \geq -\frac{\theta}{2} \|x_{k+1} - x_k\|_{\mathbf{B}_k}^2$$

$$-\frac{L_{\text{semi}}}{6} \|x_{k+1} - x_k\|_{x_k}^3 \geq -\frac{(1 + \overline{\alpha})^{3/2} L_{\text{semi}}}{6} \|x_{k+1} - x_k\|_{\mathbf{B}_k}^3 \geq -\frac{L}{6} \|x_{k+1} - x_k\|_{\mathbf{B}_k}^3$$

Combining (66) with previous inequalities, we get

$$f(x_k) - f(x_{k+1}) \geq \langle \nabla f(x_k), x_k - x_{k+1} \rangle - \frac{\theta}{2} \|x_{k+1} - x_k\|_{B_k}^2 - \frac{L}{6} \|x_{k+1} - x_k\|_{B_k}^3. $$

By adding (67), we have

$$f(x_k) - f(x_{k+1}) \geq \frac{1}{2} \langle \nabla f(x_k), x_k - x_{k+1} \rangle + \left( \frac{L}{3} - \frac{L}{6} \right) \|x_{k+1} - x_k\|_{B_k}^3. $$

Plugging in the update rule $x_{k+1} = x_k - \eta_k H_k \nabla f(x_k)$, we get

$$f(x_k) - f(x_{k+1}) \geq \frac{1}{2} \eta_k \left( \|\nabla f(x_k)\|_{B_k}^* \right)^2 + \frac{L}{6} \eta_k^3 \left( \|\nabla f(x_k)\|_{B_k}^* \right)^3. $$

Rearranging,

$$f(x_{k+1}) \leq f(x_k) - \frac{1}{2} \eta_k \left( \|\nabla f(x_k)\|_{B_k}^* \right)^2 - \frac{L}{6} \eta_k^3 \left( \|\nabla f(x_k)\|_{B_k}^* \right)^3. \qquad (68)$$

$\square$

## C   PROOF WITHOUT SEMI-STRONG SELF-CONCORDANCE

In this section, we provide a more direct alternative analysis of the CEQN algorithm under the assumption that the CEQN model upper bounds the objective function.

**Assumption 5.** *For the function $f : \mathbb{R}^d \to \mathbb{R}$ and the preconditioner schedule $\mathbf{B}_k$, there exist constants $\theta, L$ are such that and all $x, y \in \mathbb{R}^d$ holds*

$$f(y) \leq f(x_k) + \langle \nabla f(x_k), y - x_k \rangle + \frac{\theta}{2} \|y - x_k\|_{\mathbf{B}_k}^2 + \frac{L}{3} \|y - x_k\|_{\mathbf{B}_k}^3. \qquad (69)$$

This assumption can be satisfied under various conditions, or in particular:

- For $L_{semi}$-semi-strong self-concordant functions (Hanzely et al., 2022) and $\mathbf{B}_k = \nabla^2 f(x_k)$ it holds with $\theta = 1$ and $L = L_{semi}$.

- For $L_{semi}$-semi-strong self-concordant functions (Hanzely et al., 2022) and $\mathbf{B}_k$ approximating Hessian as $(1 - \underline{\alpha})\mathbf{B}_k \preceq \nabla^2 f(x_k) \preceq (1 + \overline{\alpha})\mathbf{B}_k$ it holds with $\theta = \frac{1}{1+\overline{\alpha}}$ and $L = L_{semi}\theta^{3/2}$.

  Notably, this assumption that $\mathbf{B}_k$ approximates Hessian with relative precision is standard in the analysis of Quasi-Newton methods. For (standard) self-concordant function $f$, it can be satisfied if $\mathbf{B}_k$ is chosen as Hessian at point from the neighborhood of $x_k$.

Plugging the minimizer into the upper bound leads to the following one-step decrease.

**Lemma 9.** *Quasi-Newton method with CEQN stepsize decreases functional value as*

$$f(x_{k+1}) - f(x_k) \leq -\frac{\eta_k (4 - \eta_k \theta)}{6} \|\nabla f(x_k)\|_{\mathbf{B}_k}^{*2} \tag{70}$$

$$\leq -\frac{\eta_k}{2} \|\nabla f(x_k)\|_{\mathbf{B}_k}^{*2} \tag{71}$$

$$\leq -\frac{\|\nabla f(x_k)\|_{\mathbf{B}_k}^{*2}}{2 \max\left(2\theta, \sqrt{2L\theta \|\nabla f(x_k)\|_{\mathbf{B}_k}^{*}}\right)} \tag{72}$$

$$= \begin{cases} -\frac{1}{4\theta} \|\nabla f(x_k)\|_{\mathbf{B}_k}^{*2} & \text{if } \|\nabla f(x_k)\|_{\mathbf{B}_k}^{*} \leq \frac{2\theta}{L} \\ -\frac{1}{\sqrt{8L\theta}} \|\nabla f(x_k)\|_{\mathbf{B}_k}^{*\frac{3}{2}} & \text{if } \|\nabla f(x_k)\|_{\mathbf{B}_k}^{*} \geq \frac{2\theta}{L} \end{cases} . \tag{73}$$

*Proof.* First inequality is equivalent to follows from model upperbound,

$$f(x_{k+1}) - f(x) \leq \langle \nabla f(x_k), x_{k+1} - x_k \rangle + \frac{\theta}{2} \|x_{k+1} - x_k\|_{\mathbf{B}_{x_k}}^2 + \frac{L}{3} \|x_{k+1} - x_k\|_{\mathbf{B}_{x_k}}^3 \tag{74}$$

$$= -\eta_k \|\nabla f(x_k)\|_{\mathbf{B}_{x_k}}^{*2} + \frac{\theta}{2}\eta_k^2 \|\nabla f(x_k)\|_{\mathbf{B}_{x_k}}^{*2} + \frac{L}{3}\eta_k^3 \|\nabla f(x_k)\|_{\mathbf{B}_{x_k}}^{*3} \tag{75}$$

$$= \eta_k \|\nabla f(x_k)\|_{\mathbf{B}_{x_k}}^{*2} \left(-1 + \frac{\theta}{2}\eta_k + \frac{L}{3}\eta_k^2 \|\nabla f(x_k)\|_{\mathbf{B}_{x_k}}^{*}\right), \tag{76}$$

with the choice of stepisize satisfying $1 - \theta\eta_k = L\eta_k^2 \|\nabla f(x_k)\|_{\mathbf{B}_k}^{*}$

$$= \eta_k \|\nabla f(x_k)\|_{\mathbf{B}_{x_k}}^{*2} \left(-1 + \frac{\theta}{2}\eta_k + \frac{1 - \theta\eta_k}{3}\right) \tag{77}$$

$$= \eta_k \|\nabla f(x_k)\|_{\mathbf{B}_{x_k}}^{*2} \left(-\frac{2}{3} + \frac{1}{6}\theta\eta_k\right). \tag{78}$$

The second inequality in the lemma follows from the fact that $\theta\eta_k \in (0, 1]$, and therefore $\theta\eta_k \leq 1$. The third inequality in the lemma follows from the basic manipulation of the stepsizes $\eta_k$. $\square$

Therefore, functional value decreases monotonically. As long as the gradient exponent is $3/2$, this implies $\mathcal{O}(1/k^2)$ convergence.

**Theorem 5.** *For the convex function $f : \mathbb{R}^d \to \mathbb{R}$ satisfying bounded level set assumption of the form $R \stackrel{def}{=} \max_{k \in [0,...K]} \|x_k - x^*\|_{\mathbf{B}_k} < \infty$, and the Quasi-Newton preconditioner schedule $\mathbf{B}_k$ satisfying Assumption 5, the CEQN method converges globally to point a $x_k$ such that $\|\nabla f(x_k)\|_{\mathbf{B}_k}^{*} \leq \frac{2\theta}{L}$ with the rate $\mathcal{O}(k^{-2})$.*

*Proof.* The proof is analogical to Theorem 4 of Hanzely et al. (2022).

From convexity and Cauchy-Schwarthz inequality,

$$f(x_k) - f^* \leq \langle \nabla f(x_k), x_k - x^* \rangle \leq \|\nabla f(x_k)\|_{\mathbf{B}_k}^{*} \|x_k - x^*\|_{\mathbf{B}_k} \leq R\|\nabla f(x_k)\|_{\mathbf{B}_k}^{*}. \tag{79}$$

Plugging that to Lemma 9

$$f(x_{k+1}) - f(x_k) \leq -\frac{1}{\sqrt{8L\theta}} \|\nabla f(x_k)\|_{\mathbf{B}_k^*}^{*\frac{3}{2}} \leq -\frac{1}{\sqrt{8L\theta R^3}} \left(f(x_k) - f^*\right)^{\frac{3}{2}} = -\tau \left(f(x_k) - f^*\right)^{\frac{3}{2}} \tag{80}$$

for $\tau \overset{\text{def}}{=} \frac{1}{\sqrt{8L\theta R^3}}$. Denote $\beta_k \overset{\text{def}}{=} \tau^2 \left(f(x_k) - f^*\right) \geq 0$ satisfying recurrence

$$\beta_{k+1} = \tau^2 \left(f(x_{k+1}) - f^*\right) \leq \tau^2 \left(f(x_k) - f^*\right) - \tau^3 \left(f(x_k) - f^*\right)^{3/2} = \beta_k - \beta_k^{3/2}. \tag{81}$$

Because $\beta_{k+1} \geq 0$, we have $\beta_k \leq 1$. Nesterov (2022)[Lemma A.1] shows that the sequence $\{\beta_k\}_{k=0}^{\infty}$ for $0 \leq \beta_k \leq 1$ decreases as $\mathcal{O}(k^{-2})$, so denote $c$ constant satisfying $\beta_k \leq ck^{-2}$ for all $k$ (Mishchenko (2023)[Proposition] claims that $c \approx 3$ is sufficient), then for $k$ at least

$$k \geq \sqrt{\frac{c}{\tau^2 \varepsilon}} = \sqrt{\frac{c8L\theta R^3}{\varepsilon}} = \mathcal{O}\left(\sqrt{\frac{L\theta R^3}{\varepsilon}}\right) \tag{82}$$

we have

$$f(x_k) - f^* = \frac{\beta_k}{\tau^2} \leq \frac{c}{k^2 \tau^2} \leq \varepsilon. \tag{83}$$

$\square$

# D EXPERIMENTS

Our code is available at `https://anonymous.4open.science/r/ceqn-stepsizes/`.

## D.1 EXPERIMENT DETAILS

For the $L$ parameter across all methods on `a9a` dataset, we use a logarithmically spaced grid:

$$L \in \Big\{10^{-5}, 3.16 \times 10^{-5}, 10^{-4}, 3.16 \times 10^{-4}, 10^{-3}, 3.16 \times 10^{-3}, 10^{-2},$$
$$3.16 \times 10^{-2}, 10^{-1}, 3.16 \times 10^{-1}, 1, 3.16, 10, 3.16 \times 10, 10^2, 3.16 \times 10^2, 10^3\Big\}.$$

For the $\delta$ parameter of non-adaptive Cubic Regularized Quasi-Newton (CRQN) and the $\alpha$ parameter of non-adaptive CEQN, we extend this grid to also include the value 0. For the `a9a` dataset, CEQN LSR1 used $L = 10^2$ and $\delta = 3.16 \times 10$, Adaptive CEQN reg LSR1 and dual LSR1 both used $L = 10^{-1}$, LSR1 used $L = 3.16 \times 10$, Cubic QN LSR1 used $L = 3.16 \times 10^{-5}$ and $\delta = 1$, while Adaptive Cubic QN LSR1 used $L = 3.16 \times 10^{-5}$.

For the `real-sim` dataset, we use a denser, smaller logarithmic grid:

$$L \in \Big\{10^{-5}, 2.82 \times 10^{-5}, 7.95 \times 10^{-5}, 2.24 \times 10^{-4}, 6.31 \times 10^{-4}, 1.78 \times 10^{-3},$$
$$5.02 \times 10^{-3}, 1.41 \times 10^{-2}, 3.99 \times 10^{-2}, 1.12 \times 10^{-1}, 3.17 \times 10^{-1},$$
$$8.93 \times 10^{-1}, 2.52, 7.10, 20.0\Big\}.$$

The best-performing $L$ values were: Adaptive CEQN reg LSR1 used $L = 1.12 \times 10^{-1}$, Adaptive CEQN dual LSR1 used $L = 8.93 \times 10^{-1}$, LSR1 used $L = 7.10$, Cubic QN LSR1 used $L = 7.95 \times 10^{-5}$, and Adaptive Cubic QN LSR1 used $L = 7.95 \times 10^{-5}$.

## D.2 ADDITIONAL EXPERIMENTS

In this section, we conduct all experiments on logistic regression with regularization parameter $\mu = 10^{-4}$, using the `a9a` dataset and a memory size of $m = 10$. Optimal parameters for methods in this section presented in Table 1.

|  | Adaptive CEQN reg | Adaptive CEQN dual | Classic QN | Adaptive Cubic QN |
|---|---|---|---|---|
| L-BFGS | 3.16 | $3.16 \times 10^{-1}$ | 3.16 | $3.16 \times 10^{-5}$ |
| L-BFGS history | 1 | 1 | $3.16 \times 10^2$ | $10^{-3}$ |
| SR1 | $10^{-1}$ | $10^{-1}$ | $3.16 \times 10$ | $3.16 \times 10^{-5}$ |
| SR1 history | 1 | $3.16 \times 10$ | $10^3$ | $10^{-3}$ |

Table 1: Optimal $L$ values across Hessian approximation strategies and Quasi-Newton methods.

### D.2.1 LBFGS UPDATE

In this set of experiments, we approximate the inverse Hessian $\mathbf{H}_k \approx \nabla^2 f(x_k)^{-1}$ using the limited-memory BFGS (L-BFGS) method. The approximation is based on a history of $m$ curvature pairs $(s_i, y_i)$ collected during the past optimization steps, where $s_i = x_{i+1} - x_i$ and $y_i = \nabla f(x_{i+1}) - \nabla f(x_i)$ or by sampling random directions $d_i \sim \mathcal{N}(0, I)$ and computing $s_i = d_i, y_i = \nabla^2 f(x_k)d_i$ via Hessian-vector product. These pairs are reused to construct an implicit representation of $\mathbf{H}_k$ without forming it explicitly.

We compute the product $\mathbf{H}_k \nabla f(x_k)$ using the classical two-loop recursion:

1: **Input:** Gradient $g_k = \nabla f(x_k)$, memory $\{(s_i, y_i)\}_{i=1}^m$
2: Initialize $q \leftarrow g_k$
3: **for** $i = m$ **to** 1 **do**
4: $\quad \rho_i \leftarrow 1/(y_i^\top s_i)$
5: $\quad \alpha_i \leftarrow \rho_i \cdot s_i^\top q$
6: $\quad q \leftarrow q - \alpha_i y_i$
7: Compute scalar $B_0 = \frac{y_m^\top y_m}{s_m^\top y_m}$
8: $r \leftarrow q/B_0$
9: **for** $i = 1$ **to** $m$ **do**
10: $\quad \beta \leftarrow \rho_i \cdot y_i^\top r$
11: $\quad r \leftarrow r + s_i(\alpha_i - \beta)$
12: **Return:** $r$

### D.2.2 SAMPLING VS HISTORY CURVATURE PAIRS

In this set of experiments, we compare two strategies for constructing curvature pairs $(s_i, y_i)$ used in Quasi-Newton updates. The first approach is history-based, where pairs are collected along the optimization trajectory using

$$s_i = x_{i+1} - x_i, \quad y_i = \nabla f(x_{i+1}) - \nabla f(x_i).$$

The second approach is sampling-based, in which curvature pairs are generated independently of the trajectory by drawing random directions $d_i \sim \mathcal{N}(0, I)$ and computing

$$s_i = d_i, \quad y_i = \nabla^2 f(x_k)d_i$$

via Hessian-vector products evaluated at the current iterate $x_k$.

Results are presented in Figure 3 for methods using the L-BFGS update, and in Figure 4 for those using the L-SR1 approximation.

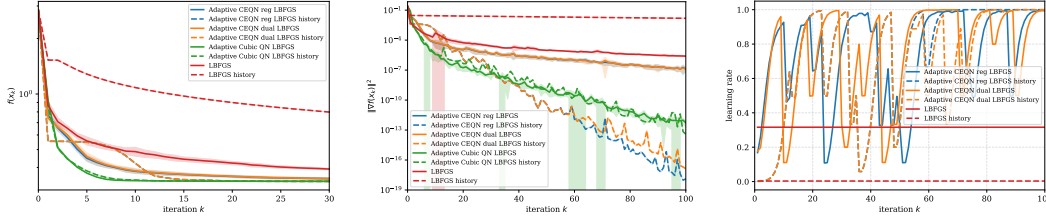

Figure 3: Comparison of different Quasi-Newton methods with BFGS updates.

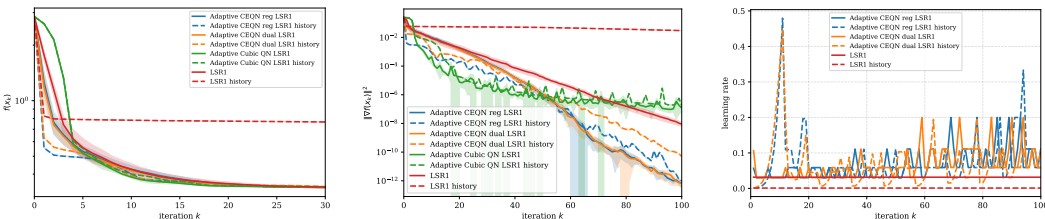

Figure 4: Comparison of different Quasi-Newton methods with SR1 updates.

### D.2.3   LSR1 vs LBFGS

In this section, we present three experiments comparing the L-SR1 and L-BFGS update rules. Specifically, we compare the two approaches using history-based curvature pairs (Figure 5), sampled curvature pairs (Figure 6), and the best-performing Quasi-Newton methods with CEQN stepsizes under each update (Figure 7).

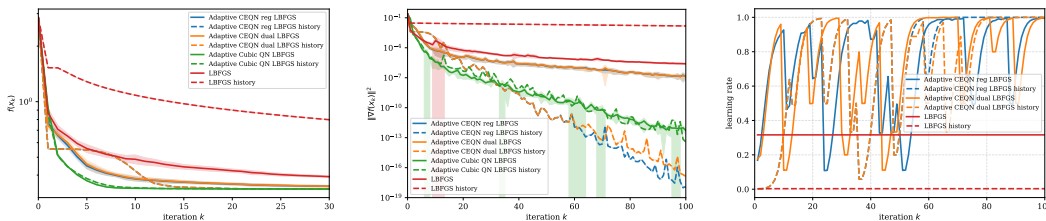

Figure 5: Comparison of LBFGS and LSR1 approximations across different Quasi-Newton methods using history-based curvature pairs.

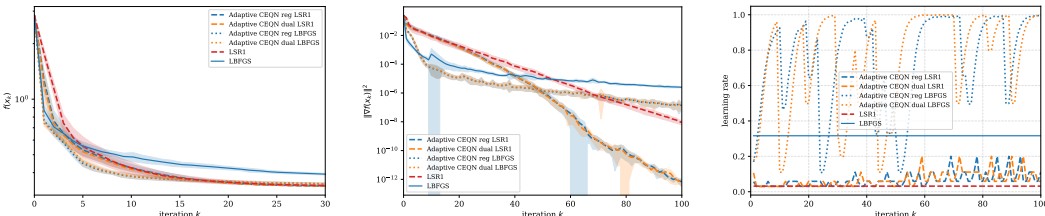

Figure 6: Comparison of LBFGS and LSR1 approximations across different Quasi-Newton methods using sampled curvature pairs.

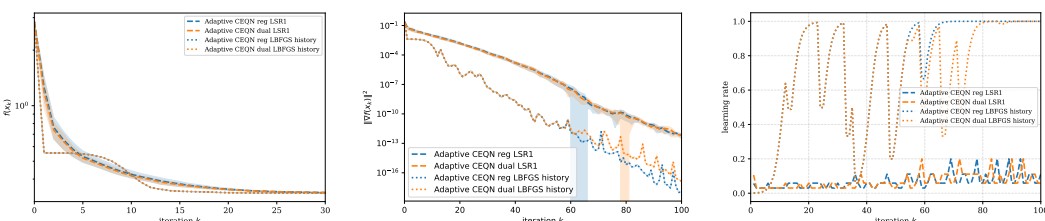

Figure 7: Comparison of the best-performing Quasi-Newton methods with adaptive CEQN stepsizes based on LBFGS and LSR1 approximations.

