# OpenReview forum: "Simple Stepsizes for Quasi-Newton Methods with Global Convergence Guarantees"
_ICLR.cc/2026/Conference — Submitted to ICLR 2026_

### Official Review · Reviewer_Rudp · 2025-10-19

**Soundness:** 2
**Presentation:** 3
**Contribution:** 2
**Rating:** 2
**Confidence:** 3

**Summary:**

This paper proposes a stepsize schedule that guarantees global convergence of the quasi-Newton methods for the general convex problem.  Their method achieves a convergence rate of $\mathcal{O}(1/k^2)$, which matches that of the accelerated gradient method.

**Strengths:**

1. This paper proposes a simple stepsize schedule for updating the quasi-Newton iteration inspired by the cubic regularized Newton method,  and it achieves global convergence for the convex problem.

2. They provide theoretical justification and empirical evidence for their proposed methods.

**Weaknesses:**

1. The main limitation of the proposed method appears to be Assumption 2, which quantifies the inexactness of the approximate Hessian with some predefined parameters $\bar{\alpha}$ and $\underset{\bar{}}{\alpha}$. However, such parameters can be quantified in existing literature (Lemma 4.1 in [1]) in the analysis of local superlinear convergence of the quasi-Newton methods for the strongly convex problems.

2. The proposed method achieves a convergence rate of $\mathcal{O}(1/k^2)$, which is comparable to that of the accelerated gradient method. However, the accelerated method is considerably simpler and does not require any (inverse) Hessian information. Therefore, the advantages of the proposed method remain unclear.

3. The name of the adaptive scheme seems misleading, as this stepsize schedule is more like a line search method.

4. There are some typos in the manuscript:
 - Line 135 and 139, I think it is $f(y)$ instead of $f(x)$
 - In the denominator of (6), should the second "+" be "-"?

**References**
- [1] Rodomanov, A., & Nesterov, Y. (2022). Rates of superlinear convergence for classical quasi-Newton methods. Mathematical Programming, 194(1), 159-190.

**Questions:**

1. Can the analysis be extended to the Broyden family without applying Assumption 2?

---

> ### Author Response · Authors · 2025-11-25
>
> Dear Reviewer Rudp,
>
> Thank you for your review and comments! Below we address your questions and comments
>
> > **Weakness 1** The main limitation of the proposed method appears to be Assumption 2, which quantifies the inexactness of the approximate Hessian with some predefined parameters $\overline{\alpha}$ and $\underline{\alpha}$. However, such parameters can be quantified in existing literature (Lemma 4.1 in [1]) in the analysis of local superlinear convergence of the quasi-Newton methods for the strongly convex problems.
>
> Thank you for raising this point. Indeed, bounds on $(\underline{\alpha}, \overline{\alpha})$ for several types of Quasi-Newton updates are known in the literature, including the result you mention. We would also like to highlight that sampling-based Quasi-Newton methods [1, Lemma 3.3] and greedy QN methods [2, 3] satisfy this assumption as well.
>
> Building on this, we introduce the adaptive CEQN scheme, which does not require any a priori knowledge of $(\underline{\alpha}, \overline{\alpha})$ and adjusts to the effective inexactness level during optimization. Results on the convergence of SR1 approximations to the true Hessian [4, Theorem 6.2] provide intuition for why the inexactness level can decrease in practice, enabling progressively larger stepsizes.
>
> [1] Berahas, Albert S., Jorge Nocedal, and Martin Takác. "A multi-batch L-BFGS method for machine learning." Advances in Neural Information Processing Systems 29 (2016).
>
> [2] Rodomanov, Anton, and Yurii Nesterov. "Greedy quasi-Newton methods with explicit superlinear convergence." SIAM Journal on Optimization 31.1 (2021): 785-811.
>
> [3] Lin, Dachao, Haishan Ye, and Zhihua Zhang. "Explicit convergence rates of greedy and random quasi-Newton methods." Journal of Machine Learning Research 23.162 (2022): 1-40.
>
> [4] Wright, Stephen, and Jorge Nocedal. "Numerical optimization." Springer Science 35.67-68 (1999): 7.
>
> > **Weakness 2** The proposed method achieves a convergence rate of $O(1/k^2)$, which is comparable to that of the accelerated gradient method. However, the accelerated method is considerably simpler and does not require any (inverse) Hessian information. Therefore, the advantages of the proposed method remain unclear.
>
> Thank you for this comment. Our goal is not to beat AGD’s $O(1/k^2)$ rate, but to obtain the same global rate for QN methods with a closed‑form, affine‑invariant stepsize that preserves the QN direction, avoiding the inner solves of cubic‑regularized QN.
> Regarding practical simplicity, AGD requires tuning two hyperparameters ([1], Algorithm 2.2.7) and is known to be sensitive to them. In contrast, CEQN uses limited-memory QN updates and therefore does not require computing the inverse Hessian. The per-iteration cost remains $O(md)$ for memory size $m$, which is comparable to AGD’s $O(d)$ in practice, especially for small $m$.
>
> Finally, in the revised version we include experiments directly comparing CEQN with AGD, and observe that CEQN outperforms AGD in practice. Please check Section 6 in the Rebuttal Revision; the new experiment is highlighted in blue.
>
> [1] Nesterov, Yurii. Lectures on Convex Optimization. Springer, 2018.
>
> > **Weakness 3** The name of the adaptive scheme seems misleading, as this stepsize schedule is more like a line search method.
>
> Thank you for the comment. We will rename the variant to “CEQN with adaptive inexactness (backtracking acceptance)”
>
> > **Weakness 4** There are some typos in the manuscript:
> Line 135 and 139, I think it is $f(y)$ instead of $f(x)$
> In the denominator of (6), should the second "+" be "-"?
>
> Thank you for pointing this out! Lines 135 and 139 are fixed (changes highlighted in blue). The denominator of (6) is correct. It corresponds to the positive root of the quadratic equation obtained from the optimality condition. Using the “−” sign would produce the non-admissible root.
>
> > **Question** Can the analysis be extended to the Broyden family without applying Assumption 2?
>
> Thank you for the question. Extending the analysis to the entire Broyden family without an assumption of the form of Assumption 2 is an interesting and nontrivial direction. The main challenge is that such an extension would require guarantees on the convergence of the approximate Hessian to the true Hessian. Establishing such guarantees would require fundamentally new analytical ideas and lies outside the scope of this work.

---

> ### Author Response · Authors · 2025-11-25
>
> ____
>
> To conclude, we believe the clarifications and added results address the concerns.
> Weaknesses 3 and 4 concern only terminology and minor typographical issues.
> The comparison with first-order methods in Weakness 2 does not contradict the novelty or the value of our contributions. The new experiment added in the rebuttal demonstrates that CEQN outperforms AGD in practice.
> Regarding Weakness 1, while we agree that Assumption 2 is restrictive, we provide explicit examples of Quasi-Newton schemes that satisfy it. Moreover, the relative inexactness model we introduce is new in the context of global second-order convergence analysis and has not been explored before.
> We hope that our clarifications, together with the theoretical guarantees and the practical performance of the proposed algorithms, will increase the value of the paper in your eyes.

---

### Official Review · Reviewer_XiRk · 2025-10-27

**Soundness:** 4
**Presentation:** 3
**Contribution:** 2
**Rating:** 4
**Confidence:** 3

**Summary:**

The paper proposes a new family of Quasi-Newton methods that combine cubic regularization and affine-invariant geometry to achieve global convergence guarantees. The core contribution is a simple explicit stepsize rule derived from the Cubically Enhanced Quasi-Newton (CEQN) framework, which yields non-asymptotic global rates of O(1/k) and, under controlled inexactness, accelerated rates of O(1/k2). The authors further develop an adaptive variant that automatically adjusts to the local level of Hessian accuracy and introduce a practical version (Algorithm 3) that allows both increasing and decreasing the regularization parameter.

The paper provides theoretical analysis under the assumption of semi-strong self-concordance, along with experiments on logistic regression tasks comparing CEQN to standard quasi-Newton and cubic-regularized baselines.

**Strengths:**

1. The analysis is carefully derived and builds on recent developments in affine-invariant Newton and quasi-Newton methods. The results are technically sound and extend prior work on cubic regularization.

2. The paper is well organized, with a logical flow from regularization-based motivation to explicit stepsize derivation and convergence proofs.

3. The explicit rule derived from the cubic model is elegant and provides a bridge between adaptive damping and regularized Newton schemes.

4. The practical adaptive scheme (Algorithm 3) is appealing; the mechanism for increasing and decreasing the inexactness level improves robustness compared to prior purely monotone approaches.

5. The experiments, though limited in scale, demonstrate consistent improvements over standard and cubic quasi-Newton baselines, validating the practical potential of the proposed stepsize.

**Weaknesses:**

1. The theoretical guarantees rely on semi-strong self-concordance, which, although broader than strong convexity, still excludes many standard objectives (e.g., generic smooth convex or non-smooth losses). The discussion would benefit from clearer intuition or examples showing when this assumption holds in practice.3

2. Experiments are restricted to small-scale convex problems. It would be valuable to see whether the method remains competitive in larger or mildly nonconvex settings.

3. The adaptive scheme shares structural similarities with inexact cubic-regularization methods (e.g., Kamzolov et al., 2023), and the novelty primarily lies in the affine-invariant reformulation and explicit stepsize rule rather than in a fundamentally new algorithmic idea.

4. Finally, although the stepsize rule is simple, the iteration complexity required to reach a target accuracy and the practical conditions under which the inexactness level can be sufficiently reduced are not fully characterized. This limits the interpretability of the convergence guarantees in practical settings.

**Questions:**

See weaknesses

---

> ### Author Response · Authors · 2025-11-25
>
> Dear Reviewer XiRk,
>
> Thank you for your constructive evaluation of our submission. We appreciate both the positive assessment of our theoretical contributions and adaptive scheme, as well as the detailed suggestions on how to strengthen the paper. Below we respond to each of the raised weaknesses point-by-point.
>
> In the revised version, we also incorporated an additional experimental comparison with Nesterov’s Accelerated Gradient Descent (AGD), directly addressing concerns regarding the practical competitiveness of our method.
>
> > **Weakness1** The theoretical guarantees rely on semi-strong self-concordance, which, although broader than strong convexity, still excludes many standard objectives (e.g., generic smooth convex or non-smooth losses). The discussion would benefit from clearer intuition or examples showing when this assumption holds in practice.
>
> Thank you for this question. Semi-strong self-concordance is a relaxation of strong self-concordance [1], which is enough to show global convergence, compared to standard self-concordance. Example 4.1 in [2] shows that any strongly convex function with Lipschitz continuous Hessian is semi-strongly self-concordant -- which includes many objectives commonly used in practice (e.g., $l2$-regularized loss in machine learning). The function $f(x)=−\log x$ is semi-strongly self-concordant but not strongly self-concordant.
>
> [1] Anton Rodomanov and Yurii Nesterov. Greedy quasi-Newton methods with explicit superlinear convergence. SIAM Journal on Optimization, 31(1):785–811, 2021.
>
> [2]  Hanzely, Slavomír, et al. "A Damped Newton Method Achieves Global $O(1/k^2)$ and Local Quadratic Convergence Rate." Advances in Neural Information Processing Systems 35 (2022): 25320-25334.
>
> > **Weakness 2** Experiments are restricted to small-scale convex problems. It would be valuable to see whether the method remains competitive in larger or mildly nonconvex settings.
>
> Thank you for this suggestion. Our work focuses on global convergence guarantees in the convex setting under relative inexactness (Assumption 2), and the experiments are designed to reflect this theoretical regime. Extending CEQN to larger-scale or mildly nonconvex problems is an interesting direction for future work, but it lies outside the scope of the present paper.
>
>
> > **Weakness 3** The adaptive scheme shares structural similarities with inexact cubic-regularization methods (e.g., Kamzolov et al., 2023), and the novelty primarily lies in the affine-invariant reformulation and explicit stepsize rule rather than in a fundamentally new algorithmic idea.
>
>
> We agree that the adaptive algorithmic design is related to the adaptive inexact cubic-Newton method from [1]. However, we extend it by adding a practical variant (Algorithm 3) and clarifying its theoretical foundation via Lemma 3.
>  Other key differences between our work and [1] include:
> • an affine-invariant formulation of both the model and the update,
> • a new relative inexactness condition (Assumption 2), and
> • a complete global non-asymptotic convergence analysis under these conditions.
>
> [1] Kamzolov, Dmitry, et al. "Accelerated adaptive cubic regularized Quasi-Newton methods."
> Journal of Optimization Theory and Applications 208.1 (2026): 1–46.
>
>
> > **Weakness 4** Finally, although the stepsize rule is simple, the iteration complexity required to reach a target accuracy and the practical conditions under which the inexactness level can be sufficiently reduced are not fully characterized. This limits the interpretability of the convergence guarantees in practical settings.
>
> Thank you for this comment. Our theoretical results provide a global non-asymptotic rate in terms of the inexactness levels in Assumption 2. The iteration complexity of CEQN depends only on the underlying Quasi-Newton update, since the proposed stepsize rule modifies only the step length, not the computational structure of the method. The practical behavior of the adaptive scheme is confirmed by our experiments, showing that adaptivity, even with backtracking, is practically beneficial and outperforms other baselines.
> Characterizing the local decay of the inexactness level $\alpha_k$ would require additional assumptions on the curvature model or on the specific quasi-Newton update used, and lies outside the scope of the present work.
> ____
> We thank the reviewer again for the constructive feedback. We hope that the clarifications provided, along with the strengthened theoretical perspective and the expanded experimental evaluation (including the AGD comparison), sufficiently address your concerns and further highlight the value of the proposed framework.

---

### Official Review · Reviewer_12wu · 2025-10-30

**Soundness:** 2
**Presentation:** 2
**Contribution:** 2
**Rating:** 2
**Confidence:** 3

**Summary:**

The work proposed an affine invariant quasi-newton algorithm with carefully chosen stepsizes for optimizing convex, semi-strongly self-concordant functions. The algorithm converges globally with rate $O(1/k^2)$ initially and transitions to $O(1/k)$. One can keep the global $O(1/k^2)$ rate if Hessian inexactness is small along the course of the algorithm. An adaptive version of the algorithm where the parameter controlling the Hessian inexactness is updated in each iteration is also proposed, and is shown to have the same global rate. Numerical validation is done against other quasi newton methods.

However, the reviewer has several major concerns on validity of the theoretical claims and practicality of the proposed algorithm, which  seem to make the paper not clearing the ICLR bar for acceptance.

**Strengths:**

1. The presentation in the manuscript is well-structured.
2. The explanation for the choice of stepsize in section 2 is very clear.
3. The literature on recent theoratical development of quasi-newton method is comprehensive.
4. The work generalizes a recent work, Affine-Invariant Cubic Newton (AICN), in allowing inexact Hessian, which makes the newly propose CEQN more versitile
5. The derivation of the stepsize rule motivated by Qubic-regularization using approximate Hessian is very nice.

**Weaknesses:**

1. **Lower bound in Assumption 2**

The paper concerns non-strictly convex objective $f$ and assumption 2 states that the Hessian $\nabla^2 f(x)$  can be under- and over-approximated by a PSD matrix $B_x$. Either the under-approximation assumption must trivialize with $\underbar{\alpha}=1$ or one must allow PSD Hessian approximation matrix $B_{x}$.

2. **On the Practical Verifiability of Assumption 2**

The entire convergence analysis of the proposed method hinges on Assumption 2, the relative inexactness condition $(1-\underline{\alpha})B_{x} \le \nabla^{2}f(x) \le (1+\overline{\alpha})B_{x}$. The resulting convergence rates in Theorem 1 depend directly on the inexactness parameters $\underline{\alpha}$ and $\overline{\alpha}$. However, the paper provides no practical, computationally feasible method to construct or verify an approximation $B_x$ that is guaranteed to satisfy this assumption for known (or even bounded) values of $\underline{\alpha}$ and $\overline{\alpha}$. While the experiments employ standard methods like L-BFGS, these are recursive methods that do not, in general, come with a priori guarantees on their spectral relationship to the true Hessian $\nabla^{2}f(x)$. It is unclear how these practical methods connect back to the foundational assumption of the theory. This creates a significant gap between the theory and practice. For instance, the parameter selection for the non-adaptive algorithm (e.g., $\theta \ge 1+\overline{\alpha}$) is impossible to implement without a priori knowledge of $\overline{\alpha}$. The adaptive method  attempts to find this level, but it does so by reacting to a failed condition rather than by verifying the assumption itself. The analysis lacks a crucial component: a practical algorithm to compute a $B_x$ that demonstrably satisfies the conditions upon which the theory is built. The authors do discuss in the appendix (L1359) that one may satisfy such assumption if $B_k$ is chosen as the exact hessian at a point near $x_k$, but assuming that we can compute the exact Hessian defies the purpose of Newton-type or quasi-Newton method.

3. **Trajectory-dependent constants**

This is the biggest concern that the reviewer has on the theoretical contribution of this work. The analysis in Theorem 1 presents a convergence rate that depends on a quantity $D=\max_{k\in[0;K+1]} ||x_{k}-x_{*}||_{B_k}$.

This makes the stated convergence bound circular. A standard, a priori complexity bound should be expressed in terms of initial problem parameters (e.g., $||x_0 - x_*||$ and function properties) and should allow one to predict the number of iterations $K$ required to achieve a target accuracy $\epsilon$. As written, the bound in Theorem 1 cannot be used for this purpose, since the value of $D$ is unknown until after the algorithm has already run for $K$ steps. While Lemma 2 establishes a valuable descent property on the function value, this alone is insufficient to guarantee a uniform (i.e., $K$-independent) bound on $D$, especially as the norm $|| \cdot ||_{B_k}$ changes at each iteration. This reliance on a posteriori quantities seems to be a foundational part of the analysis, as it also appears in the definition of $\overline{D}$ for the adaptive method in Theorem 2 4 and again as $R$ in the alternative analysis of Theorem 5. The theoretical claims would be significantly strengthened if the authors could provide true a priori bounds based on a more standard analysis, such as by assuming a bounded initial level set.

4. **On the Condition for the $O(k^{-2})$ Rate (Corollary 2)**

The paper claims that an $\mathcal{O}(k^{-2})$ rate is achievable if the inexactness satisfies the condition $\alpha_k \le L||x_{k+1}-x_{k}||_{B_k}$.

However, this condition is inherently a posteriori. It cannot be guaranteed before taking a step, as the right-hand side of the inequality depends on the next iterate $x_{k+1}$, which is itself a result of the step taken. The choice of $\alpha_k$ influences the step, which in turn determines if the condition is met.Therefore, one cannot enforce this condition to guarantee the accelerated rate. One can only check after the fact if the condition was satisfied at each iteration. The paper itself notes that the condition is "verifiable in practice"2, which reinforces this point. As such, Corollary 2 does not provide a constructive method to achieve the $\mathcal{O}(k^{-2})$ rate, but rather identifies a specific, non-enforceable scenario under which it occurs.


5. **QN vs. Newton-type**

The authors refer to their proposed method as a "Quasi-Newton" (QN) method. While the term is used broadly, in optimization literature, "Quasi-Newton" most often refers to a specific class of methods (like BFGS, DFP, or SR1) where the Hessian approximation $B_k$ (or its inverse $H_k$) is constructed recursively using iterate and gradient differences, satisfying a secant equation. It would be more appropriate to name the proposed method as "Newton-type" rather than QN. I was initially excited that this work would have established the global convergence guarantee for QN methods in the literature, but the proposed algorithm was rather of a Newton-type.

6. **Comparison to Existing Work**

In the introduction, the authors state that global convergence rate for Newton-type or quasi-newton method is not known in the literature --- "Despite all of the interest, even nowadays, many classical Quasi-Newton methods still lack non-asymptotic global convergence guarantees. Only recently global non-asymptotic convergence guarantees with explicit rates were established for BFGS in the strongly convex setting for specific line search procedures". Their primary goal is  "to guarantee global non-asymptotic convergence guarantees for classical Quasi-Newton methods for non-strongly convex functions." Precisely this has been done in a recent work (Duan, Lyu 2025; arXiv:2509.21684) on a different but related algorithm, which relaxes Mishchenko's globally regularized Newton (Mishchenko, 2021; arXiv:2112.02089) with overestimated Hessian approximation with also a very simple stepsize rule. This work provides some practical randomized Hessian overestimation procedure, obtains $O(1/k)\rightarrow O(1/k^2)$ global convergence rate (similar to Thm. 1), and local convergence analysis in rank-deficient landscape. Another related work is (Doikov, Stich, Jaggi, 2024; arxiv: 2402.04843), where stronger convervgence guarantees are provided assuming some global knowledge of Hessian spectrum. Certainly the author's goal has been considered in the literature and very relevant results are already obtained. The author's should discuss how their result compares to this work and modify the introduction accordingly.


7. **Notational inconsistencies**

Some inconsistancies in notations. For example, it seems to me that some of the $L$'s between line 163 and 170 should be $L_2$ instead, and $\eta$ on line 201 should be $\eta_k$. At line 116, given fixed $\mathbf{B}$, one defines the operator norm of another matrix $\mathbf{A}$ as $$\|\mathbf{A}\|_{op}=\sup_{y\in\mathbb{R^d}}\frac{\|\mathbf{A}y\|_{x}^{*}}{\|y\|_x},$$ but the right hand side does not depend on $\mathbf{B}.$ Maybe the author(s) want to define the operator norm for a fixed reference point $x$ instead?

7. **Lack of local convergence analysis.**

One benefit of Newton-type methods is that one can still have superlinear local convergence without processing the full Hessian. It seems unlikely that (adapted) CEQN can have superlinear local convergence since it is using Hessian approximation from limited memory and the stepsize is not going to 1. In this case, it seems the benefit of using CEQN versus AGD is unclear, given that AGD also have global rate $O(1/k^2).$




**Minor comments:**

L108:  holds --> it holds that

L198: an stepsise --> a stepwise

L201: \eta-->\eta_k

L201: The exposition here is rough. The expression $\Vert h_k \Vert_{B_k}=\eta_k \Vert \nabla f(x) \Vert _{B_k}$ is plugged back to the definition of $\eta_k$ to yield the quadratic equation for $\eta_k$.

**Questions:**

1. Can author(s) clarify the definition of the operator norm of a matrix (L115): whether it is with respect to a fix point $x$ or a fix different matrix $B$? Is it a typo that the norms in the definition there should be subscripted with $B$? And in either case, what is the fixed point (or matrix) when the operator norm is envoked in Assumption 1?

2. In Theorem 1, can author(s) upper bound $D$ by some quantity independent of $K$? Otherwise, it is not transparent that the bound in Theorem 1 gives the desired non-asympototic global rate $O(1/k^2)$ or $O(1/k)$. Similarly for the quantity $\overline{D}$ and $\alpha_K$ in Theorem 2.

3. Can author(s) provide some numerical experiments of (adaped)CEQN against AGD? I also wonder how much does allowing inexact Hessian help with speeding up the computation. Can author(s) also provide some experiments against AICN (https://arxiv.org/pdf/2211.00140)?

---

> ### Author Response · Authors · 2025-11-25
>
> Dear Reviewer 12wu,
>
> Thank you for your thorough and detailed evaluation of our submission. We appreciate the substantial time you invested into analyzing the technical aspects of the paper. Below we respond to each of your comments in detail.
>
>
>
> > **Weakness1** The paper concerns non-strictly convex objective f  and assumption 2 states that the Hessian \nabla^2 f(x)  can be under- and over-approximated by a PSD matrix . Either the under-approximation assumption must trivialize with \undeline{\alpha} = 1  or one must allow PSD Hessian approximation matrix B_x.
>
> Thank you for your comment. Assumption 2 explicitly allows $\underline{\alpha} = 1$.
>
>
> > **Weakness 2** On the Practical Verifiability of Assumption 2
> The entire convergence analysis of the proposed method hinges on Assumption 2, the relative inexactness condition $(1-\underline{\alpha})B_{x} \le \nabla^{2}f(x) \le (1+\overline{\alpha})B_{x}$. The resulting convergence rates in Theorem 1 depend directly on the inexactness parameters $\underline{\alpha}$ and $\overline{\alpha}$. However, the paper provides no practical, computationally feasible method to construct or verify an approximation $B_x$  that is guaranteed to satisfy this assumption for known (or even bounded) values of $\underline{\alpha}$ and $\overline{\alpha}$. While the experiments employ standard methods like L-BFGS, these are recursive methods that do not, in general, come with a priori guarantees on their spectral relationship to the true Hessian $\nabla^{2}f(x)$. It is unclear how these practical methods connect back to the foundational assumption of the theory. This creates a significant gap between the theory and practice…The analysis lacks a crucial component: a practical algorithm to compute a  that demonstrably satisfies the conditions upon which the theory is built. The authors do discuss in the appendix (L1359) that one may satisfy such assumption if  is chosen as the exact hessian at a point near , but assuming that we can compute the exact Hessian defies the purpose of Newton-type or quasi-Newton method.
>
> Thank you for raising this concern.
>
> Several Hessian approximations satisfy Assumption 2. First, sampling-based quasi-Newton methods. In this setting, the accuracy can be improved by increasing the number of sampled pairs. For BFGS, it is known that for strongly convex functions with Lipschitz-continuous gradients, the iterates satisfy $c_1 I \preceq B \preceq c_2 I$ [Lemma 3.3, 1]. Combining this with $\mu$-strong convexity and $L_1$​-smoothness of $\nabla f(x)$, one can derive $\frac{\mu}{c_2} B \preceq \nabla^2 f \preceq \frac{L_1}{c_1} B$, which gives explicit values for $(\underline \alpha, \overline \alpha)$ in Assumption 2. We acknowledge, however, that these bounds can be conservative in practice — which is exactly why we propose the adaptive scheme that adapts to the effective inexactness level during optimization. Moreover, Assumption 2 is satisfied for Greedy QN methods [2, 3] and slightly modified classic QN methods [4]. Additionally, there are results on the convergence of SR1 approximations to the true Hessian [5, Theorem 6.2 ]. Finally, we emphasize that our theoretical framework is general and applies to any preconditioner satisfying Assumption 2. For example, in stochastic settings, the quality of the preconditioner can often be directly controlled via sampling mechanisms.
>
> Most importantly, our adaptive schemes never requires knowing $(\underline{\alpha}, \overline{\alpha})$. Instead, Algorithm 2 verifies feasibility of the current step and increases $\theta$ only when the theoretical decrease condition is violated.
>
>
>
> [1] Berahas, Albert S., Jorge Nocedal, and Martin Takác. "A multi-batch L-BFGS method for machine learning." Advances in Neural Information Processing Systems 29 (2016).
>
> [2] Rodomanov, Anton, and Yurii Nesterov. "Greedy quasi-Newton methods with explicit superlinear convergence." SIAM Journal on Optimization 31.1 (2021): 785-811.
>
> [3] Lin, Dachao, Haishan Ye, and Zhihua Zhang. "Explicit convergence rates of greedy and random quasi-Newton methods." Journal of Machine Learning Research 23.162 (2022): 1-40.
>
> [4] Rodomanov, Anton, and Yurii Nesterov. "Rates of superlinear convergence for classical quasi-Newton methods." Mathematical Programming 194.1 (2022): 159-190.
>
> [5] Wright, Stephen, and Jorge Nocedal. "Numerical optimization." Springer Science 35.67-68 (1999): 7.

---

> ### Author Response · Authors · 2025-11-25
>
> > **Weakness 2**
>  …For instance, the parameter selection for the non-adaptive algorithm (e.g.,
> $\theta \ge 1+\overline{\alpha}$) is impossible to implement without a priori knowledge of $\overline{\alpha}$. The adaptive method attempts to find this level, but it does so by reacting to a failed condition rather than by verifying the assumption itself…
>
> You are correct that the non-adaptive variant requires $\theta \geq 1 + \overline{\alpha}$, which is not available in practice. This is exactly why Algorithm 2 is introduced. The adaptive scheme does not attempt to estimate $\overline{\alpha}$ directly; instead, it verifies the feasibility of the current step and adjusts $\theta$ only when the theoretical sufficient condition is violated. This “backtracking-style” mechanism is standard in adaptive methods, where one enforces the conditions required by the theory without explicitly estimating the underlying model parameters. Our empirical results show that this leads to robust practical performance.
>
> > **Weakness 3** Trajectory-dependent constants.
> > This is the biggest concern that the reviewer has on the theoretical contribution of this work. The analysis in Theorem 1 presents a convergence rate that depends on a quantity $D=\max \limits_{k \in [0;K+1]} ||x_{k}-x^{*}||_{B}$.
>
> > This makes the stated convergence bound circular. A standard, a priori complexity bound should be expressed in terms of initial problem parameters (e.g., $ ||x_0 - x_*|| $ and function properties) and should allow one to predict the number of iterations $K$  required to achieve a target accuracy $\epsilon$. As written, the bound in Theorem 1 cannot be used for this purpose, since the value of $D$  is unknown until after the algorithm has already run for $K$  steps. While Lemma 2 establishes a valuable descent property on the function value, this alone is insufficient to guarantee a uniform (i.e.,$K$ -independent) bound on $D$, especially as the norm  $||x_0 - x_*||$ changes at each iteration. This reliance on a posteriori quantities seems to be a foundational part of the analysis, as it also appears in the definition of $\overline{D}$  for the adaptive method in Theorem 2 4 and again as $R$  in the alternative analysis of Theorem 5. The theoretical claims would be significantly strengthened if the authors could provide true a priori bounds based on a more standard analysis, such as by assuming a bounded initial level set.
>
> > **Question 2** In Theorem 1, can author(s) upper bound $D$  by some quantity independent of $K$? Otherwise, it is not transparent that the bound in Theorem 1 gives the desired non-asympototic global rate $O(1/k^2)$  or $O(1/k)$. Similarly for the quantity $\overline{D}$ and $\alpha_K$ in Theorem 2.
>
> Thank you for this question. We have added Remark 1 and Remark 2 in the main text (both highlighted in blue), together with full proofs in the appendix. These results show that both $D$ and \overline{D} are upper-bounded solely by problem constants, and do not depend on $K$. Finally, $\alpha_K$ is automatically bounded above by the maximal relative inexactness level over the iterates, which is again a problem-dependent constant.

---

> ### Author Response · Authors · 2025-11-25
>
> > **Weakness 4** On the Condition for the  Rate (Corollary 2)
> The paper claims that an $O(1/k^2)$  rate is achievable if the inexactness satisfies the condition $\alpha_k \leq L\|x_{k+1} - x_k\|_{{B_{k}}}$.
> However, this condition is inherently a posteriori. It cannot be guaranteed before taking a step, as the right-hand side of the inequality depends on the next iterate $x_{k+1}$, which is itself a result of the step taken. The choice of $\alpha_k$ influences the step, which in turn determines if the condition is met.Therefore, one cannot enforce this condition to guarantee the accelerated rate. One can only check after the fact if the condition was satisfied at each iteration. The paper itself notes that the condition is "verifiable in practice"2, which reinforces this point. As such, Corollary 2 does not provide a constructive method to achieve the $O(k^{-2})$  rate, but rather identifies a specific, non-enforceable scenario under which it occurs.
>
>
> Thank you for your comment. We agree that this condition can indeed be verified only after the step is taken. This is fully intentional: it is a standard a posteriori sufficient condition, commonly used in optimization methods. The condition is verifiable and practical, and this does not introduce any inconsistency.
> A posteriori acceptance tests are widespread in modern optimization. Many widely-used methods take a step, verify a sufficient condition, and adjust parameters if needed. Just to name a few:
> - Gradient descent with backtracking line search (Algorithm 9.2 in [1])
> - Bisection-based stepsize selection in near-optimal and optimal second-order and tensor methods. For example,  Monteiro–Svaiter A-HPE ( Section 6 in [2]) Optimal tensor methods (Algorithm 3 in [3])
> - Backtracking line search in globally convergent Quasi-Newton methods (Subroutine 1 and Algorithm 1 in [4])
> Such conditions are especially common in methods with inexact derivatives. For instance, Conditions 4.1 and 4.2 in [5], as well as the sufficient inexactness conditions in Theorem 3.5 of [6], are also verified a posteriori.
> Therefore, Corollary 2 provides a valid sufficient condition under which the accelerated $O(1/k^2)$ rate holds. The fact that this condition is checked after computing the candidate step is entirely standard and does not reduce its usefulness or practicality.
>
>
>
> [1] Boyd, Stephen, and Lieven Vandenberghe. Convex optimization. Cambridge university press, 2004.
>
> [2] Monteiro, Renato DC, and Benar Fux Svaiter. "An accelerated hybrid proximal extragradient method for convex optimization and its implications to second-order methods." SIAM Journal on Optimization 23.2 (2013): 1092-1125.
>
> [3] Kovalev, Dmitry, and Alexander Gasnikov. "The first optimal acceleration of high-order methods in smooth convex optimization." Advances in Neural Information Processing Systems 35 (2022): 35339-35351.
>
> [4] Jiang, Ruichen, Qiujiang Jin, and Aryan Mokhtari. "Online learning guided curvature approximation: A quasi-newton method with global non-asymptotic superlinear convergence." The Thirty Sixth Annual Conference on Learning Theory. PMLR, 2023.
>
> [5] Lin, Tianyi, Panayotis Mertikopoulos, and Michael I. Jordan. "Explicit second-order min-max optimization methods with optimal convergence guarantee." arXiv preprint arXiv:2210.12860 (2022).
>
> [6] Agafonov, Artem, et al. "Exploring jacobian inexactness in second-order methods for variational inequalities: lower bounds, optimal algorithms and quasi-newton approximations." Advances in Neural Information Processing Systems 37 (2024): 115816-115860.

---

> ### Author Response · Authors · 2025-11-25
>
> > **Weakness 5** QN vs. Newton-type
> The authors refer to their proposed method as a "Quasi-Newton" (QN) method. While the term is used broadly, in optimization literature, "Quasi-Newton" most often refers to a specific class of methods (like BFGS, DFP, or SR1) where the Hessian approximation  (or its inverse ) is constructed recursively using iterate and gradient differences, satisfying a secant equation. It would be more appropriate to name the proposed method as "Newton-type" rather than QN. I was initially excited that this work would have established the global convergence guarantee for QN methods in the literature, but the proposed algorithm was rather of a Newton-type.
>
> While our theoretical results apply to any sequence of preconditioners satisfying Assumption 2, in this work we intentionally focus on QN methods, as they are arguably the most practical and widely adopted approach for building approximate second-order information in large-scale optimization — especially when computing or storing full Hessians is infeasible. The paper is motivated by improving and analyzing QN methods in the context of cubic regularization, and we believe that this clear focus makes the contribution both technically rigorous and practically relevant. Therefore, although our analysis indeed covers general Newton-type schemes, we deliberately keep the framing in terms of QN methods, which are the primary target and motivation of this work.
>
>
>
> > **Weakness 6** Comparison to Existing Work
> In the introduction, the authors state that global convergence rate for Newton-type or quasi-newton method is not known in the literature --- "Despite all of the interest, even nowadays, many classical Quasi-Newton methods still lack non-asymptotic global convergence guarantees. Only recently global non-asymptotic convergence guarantees with explicit rates were established for BFGS in the strongly convex setting for specific line search procedures". Their primary goal is "to guarantee global non-asymptotic convergence guarantees for classical Quasi-Newton methods for non-strongly convex functions." Precisely this has been done in a recent work (Duan, Lyu 2025; arXiv:2509.21684) on a different but related algorithm, which relaxes Mishchenko's globally regularized Newton (Mishchenko, 2021; arXiv:2112.02089) with overestimated Hessian approximation with also a very simple stepsize rule. This work provides some practical randomized Hessian overestimation procedure, obtains  global convergence rate (similar to Thm. 1), and local convergence analysis in rank-deficient landscape. Another related work is (Doikov, Stich, Jaggi, 2024; arxiv: 2402.04843), where stronger convervgence guarantees are provided assuming some global knowledge of Hessian spectrum. Certainly the author's goal has been considered in the literature and very relevant results are already obtained. The author's should discuss how their result compares to this work and modify the introduction accordingly.
>
> Thank you for highlighting these additional references.
> The preprint by Duan and Lyu (2025, arXiv:2509.21684) appeared online after the conference submission deadline and was not available at the time of writing.
>
> Regarding Doikov, Stich, and Jaggi (2024, arXiv:2402.04843), their analysis focuses on spectral preconditioners and primarily targets nonconvex objectives. Their guarantees rely on global bounds on the Hessian spectrum, which is a substantially different setting from ours.
> For these reasons, we view both works as related but not addressing the same problem or algorithmic family. Our contribution remains distinct.
>
>
> > **Weakness 7** Notational inconsistencies
> Some inconsistancies in notations. For example, it seems to me that some of the $L$'s between line 163 and 170 should be $L_2$ instead, and $\eta$ on line 201 should be $\eta_k$. At line 116, given fixed $B$, one defines the operator norm of another matrix  as $$|\mathbf{A}|{op}=\sup{y\in\mathbb{R^d}}\frac{|\mathbf{A}y|_{x}^{*}}{|y|_x},$$ but the right hand side does not depend on $B$. Maybe the author(s) want to define the operator norm for a fixed reference point instead?
> > **Question 1** Can author(s) clarify the definition of the operator norm of a matrix (L115): whether it is with respect to a fix point $x$  or a fix different matrix $B$? Is it a typo that the norms in the definition there should be subscripted with $B$? And in either case, what is the fixed point (or matrix) when the operator norm is envoked in Assumption 1?
> > **Minor changes**
>
> Thank you for pointing this out. We have corrected these notational inconsistencies and updated the affected expressions in the resubmitted version. For clarity, the operator norm in line 115 is defined with respect to the Hessian-induced local norm at a fixed reference point $x$.

---

> ### Author Response · Authors · 2025-11-25
>
> > **Weakness 8** Lack of local convergence analysis.
> One benefit of Newton-type methods is that one can still have superlinear local convergence without processing the full Hessian. It seems unlikely that (adapted) CEQN can have superlinear local convergence since it is using Hessian approximation from limited memory and the stepsize is not going to 1.
>
> Thank you for raising this point. In this work we focus on global non-asymptotic convergence guarantees under relative inexactness. Extending the theory to the strongly convex setting and studying local convergence is an interesting direction for future work.
>
>
> > **Weakness 8**In this case, it seems the benefit of using CEQN versus AGD is unclear, given that AGD also have global rate $O(1/k^2)$.
> > **Question 3** Can author(s) provide some numerical experiments of (adaped)CEQN against AGD? I also wonder how much does allowing inexact Hessian help with speeding up the computation. Can author(s) also provide some experiments against AICN (https://arxiv.org/pdf/2211.00140)?
>
> Thank you for raising this point. We have added a comparison between CEQN and Nesterov’s Accelerated Gradient Descent (AGD). The new experiments are reported in Figure 2 of the revised manuscript, where CEQN–LSR1 and CEQN–LBFGS consistently outperform AGD in both iteration count and wall-clock time.
> While our theory does not claim that CEQN outperforms AG, our goal is different: we provide explicit, affine-invariant stepsizes for quasi-Newton updates with global guarantees under convexity, and we show that when the relative inexactness is controlled the method attains $O(1/k^2)$—matching AGD’s global rate.
> Regarding AICN, we are currently running additional experiments. However, we note that AICN requires matrix inversion (or solving linear system) at each iteration (naively $O(d^3)$), similar to a Newton method. In contrast, CEQN with limited-memory QN updates constructs an inverse Hessian approximation using curvature pairs with $O(md)$ complexity. This makes the comparison less central to the practical regime our paper focuses on. We plan to include the AICN comparison in the camera-ready version.
>
>
> ____
>
> We appreciate the reviewer’s careful reading of the paper. We hope that the clarifications above address all raised concerns. We would like to stress that we do not view our adaptivity mechanism or the a-posteriori sufficient condition for achieving the exact $O(1/k^2)$ rate as limitations. Both are standard in modern optimization, and our experiments — including the newly added comparison with AGD — demonstrate the practical relevance and robustness of the proposed method. While we agree that proposing a new Hessian-approximation scheme might strengthen the paper, we explicitly point to several classes of updates that satisfy our assumptions in theory and work well in practice. We thank the reviewer again for the thoughtful feedback.

---

### Official Review · Reviewer_tV4b · 2025-10-31

**Soundness:** 3
**Presentation:** 3
**Contribution:** 2
**Rating:** 4
**Confidence:** 5

**Summary:**

This paper introduces a new stepsize scheme for Quasi-Newton (QN) methods—Cubically Enhanced Quasi-Newton (CEQN)—which yields global non-asymptotic convergence guarantees for convex optimization problems. Classical QN methods (like BFGS and SR1) are efficient locally but lack explicit global convergence rates without line search or strong convexity assumptions. CEQN bridges this gap by: 1. Deriving a simple analytic stepsize from a cubic-regularization viewpoint. 2. Achieving O(1/k) global convergence rate and O(1/k²) under controlled Hessian inexactness. 3. Introducing an adaptive variant that automatically adjusts to Hessian approximation quality. 4. Demonstrating superior empirical performance over standard QN and cubic-regularized baselines on logistic regression tasks.

**Strengths:**

This paper cleverly connects cubic regularization and QN updates through a unified stepsize formula, which is a theoretical contribution. This expression mirrors the implicit damping of cubic regularization while retaining the simplicity of standard QN updates. The authors prove explicit O(1/k²) rates under verifiable conditions on Hessian inexactness—an improvement over prior QN analyses, which often only show asymptotic convergence.

This paper connect the quasi-Newton method's property of the affine-invariant analysis. The affine-invariance property is preserved both algorithmically and theoretically, aligning the work with Newton-type methods’ geometric robustness. The authors also proposed two adaptive mechanism: 1. the adaptive CEQN (Algorithm 2) adjusts the inexactness level αₖ dynamically, maintaining global convergence while avoiding manual stepsize tuning. 2. the practical variant (Algorithm 3) introduces both increasing and decreasing mechanisms, balancing theoretical rigor and empirical efficiency.

The authors also conduct numerical experiments on a9a and real-sim logistic regression showing clear advantages in both iteration count and wall-clock time. The adaptive CEQN variants outperform classical LSR1 and cubic QN in gradient convergence and loss reduction. The empirical results are consistent with the theoretical analysis.

**Weaknesses:**

There are three major weaknesses:

1. the step size depends on the constant L, which is dependent on the parameters of the objective function. These parameters are in general hard to estimate. Therefore, this make this step size difficult to implement in practice.

2. The assumption 2 is too strong. In general, we can't assume that the Hessian approximation matrix is close to the exact Hessian. This should be proved. Moreover, the inexact Hessian parameters need to be estimated with explicit lower and upper bounds. And the final convergence rates need to be dependent on these parameters.

3. The most important property of the quasi-Newton method is the superlinear convergence rate. The authors need to prove that with the additional strong convexity assumption, the proposed method can also achieve the superlinear convergence. Otherwise, there is no advantages over the gradient and accelerated gradient method. These methods also reach the rate of 1/k and 1/k^2 and the computational cost per iteration of these methods are smaller than quasi-Newton method. Without the superlinear convergence analysis, the proposed method didn't have advantages over the first-order methods.

**Questions:**

There are no other questions.

---

> ### Author Response · Authors · 2025-11-25
>
> Dear Reviewer tV4b,
>
> Thank you for providing constructive and technically detailed feedback. We greatly appreciate your positive assessment of both the theoretical contribution and practical relevance of our work. Below, we address each of your three main concerns in detail
>
> > **Weakness1** the step size depends on the constant L, which is dependent on the parameters of the objective function. These parameters are in general hard to estimate. Therefore, this make this step size difficult to implement in practice.
>
> We agree that the constant $L$ is unknown a priori and hard to estimate. In our adaptive CEQN, $L$ is the only scalar we tune (Sec. 5; App. D), analogous to gradient descent where the ideal learning rate $1/L_1$  ​ is based on an unknown gradient-Lipschitz constant $L_1$, so the learning rate is tuned in practice.
>
> > **Weakness 2** The assumption 2 is too strong. In general, we can't assume that the Hessian approximation matrix is close to the exact Hessian. This should be proved. Moreover, the inexact Hessian parameters need to be estimated with explicit lower and upper bounds. And the final convergence rates need to be dependent on these parameters.
>
> Thank you for raising this concern. Assumption 2 is a general relative inexactness model used for the analysis. Our guarantees hold for any matrix $B_x$ that satisfies it.
> Assumption 2 does not require $B_x$ to be highly accurate; it only bounds the relative distortion between $B_x$ and $\nabla^2 f(x)$. Algorithm 1 does not require a priori bounds on $(\underline{\alpha}, \overline{\alpha})$.  The convergence rate bounds in our theory explicitly depend on $(\underline{\alpha}, \overline{\alpha})$; see Theorem 1 for the stated global guarantee and its parameter requirements.
> Algorithms 2–3 are fully adaptive and do not use these parameters. The theory only requires that some (possibly unknown) $(\underline{\alpha}, \overline{\alpha})$ exist.
> Several Hessian approximations satisfy Assumption 2. First, sampling-based quasi-Newton methods. In this setting, the accuracy can be improved by increasing the number of sampled pairs. For BFGS, it is known that for strongly convex functions with Lipschitz-continuous gradients, the iterates satisfy $c_1 I \preceq B \preceq c_2 I$ [Lemma 3.3, 1]. Combining this with $\mu$-strong convexity and $L_1$​-smoothness of $\nabla f(x)$, one can derive $\frac{\mu}{c_2} B \preceq \nabla^2 f \preceq \frac{L_1}{c_1} B$, which gives explicit values for $(\underline \alpha, \overline \alpha)$ in Assumption 2. We acknowledge, however, that these bounds can be conservative in practice — which is exactly why we propose the adaptive scheme that adapts to the effective inexactness level during optimization. Moreover, Assumption 2 is satisfied for Greedy QN methods [2, 3] and slightly modified classic QN methods [4]. Additionally, there are results on the convergence of SR1 approximations to the true Hessian [5, Theorem 6.2 ].
>
> [1] Berahas, Albert S., Jorge Nocedal, and Martin Takác. "A multi-batch L-BFGS method for machine learning." Advances in Neural Information Processing Systems 29 (2016).
>
> [2] Rodomanov, Anton, and Yurii Nesterov. "Greedy quasi-Newton methods with explicit superlinear convergence." SIAM Journal on Optimization 31.1 (2021): 785-811.
>
> [3] Lin, Dachao, Haishan Ye, and Zhihua Zhang. "Explicit convergence rates of greedy and random quasi-Newton methods." Journal of Machine Learning Research 23.162 (2022): 1-40.
>
> [4] Rodomanov, Anton, and Yurii Nesterov. "Rates of superlinear convergence for classical quasi-Newton methods." Mathematical Programming 194.1 (2022): 159-190.
>
> [5] Wright, Stephen, and Jorge Nocedal. "Numerical optimization." Springer Science 35.67-68 (1999): 7.

---

> ### Author Response · Authors · 2025-11-25
>
> > **Weakness 3** The most important property of the quasi-Newton method is the superlinear convergence rate. The authors need to prove that with the additional strong convexity assumption, the proposed method can also achieve the superlinear convergence. Otherwise, there is no advantages over the gradient and accelerated gradient method. These methods also reach the rate of 1/k and 1/k^2 and the computational cost per iteration of these methods are smaller than quasi-Newton method. Without the superlinear convergence analysis, the proposed method didn't have advantages over the first-order methods.
>
> Thank you for this suggestion. Superlinear local convergence is indeed an important property of classical Quasi-Newton methods under strong convexity. However, in this work we focus on global convergence guarantees under relative inexactness (Assumption 2). Extending our framework to the strongly convex setting and establishing local superlinear convergence is an interesting direction for future work.
>
>
> _____
>
> We hope that our clarifications address your concerns. We hope that the above clarifications address your concerns. Our goal is not to replace or compete with the classical superlinear convergence theory of Quasi-Newton methods, but to bridge the gap between practical QN updates and explicit global convergence guarantees.
> In the revised version, we have also included an additional experiment comparing CEQN against Accelerated Gradient Descent (AGD). Despite AGD having two tuned hyperparameters and using acceleration, it is consistently slower than CEQN in both iteration count and wall-clock time.
> While explicit lower/upper bounds on Hessian inexactness may be hard to obtain in general, our adaptive CEQN variants demonstrate strong practical robustness: they automatically adapt to the effective inexactness level without requiring any prior knowledge of these parameters.
> We appreciate your thoughtful comments and believe that the revisions and explanations above substantially strengthen the clarity, practical framing, and theoretical positioning of the paper. Thank you again for your time and valuable feedback.

---

### Author Response · Authors · 2025-11-25

Dear Reviewers,

Thank you for your feedback. We have uploaded a revised version of the manuscript, where all updates and newly added clarifications are highlighted in blue.

---

### Meta-Review · Area_Chair_yArC · 2025-12-15

**Summary:**

This paper presents an adaptive quasi-Newton method with the global convergence of $O(1/k^2)$. The main weaknesses include the convergence of proposed method is no better than accelerated gradient descent and there is no superlinear local convergence rates. Additionally, Assumption 2 is unreasonable.  In my opinion, we should prove equation (9) by the property of the objective function and the algorithm, rather than suppose it holds directly. Since above concerns have not been addressed at rebuttal stage, I recommend rejection.

**Reviewer Concerns:**

Some main concerns:
1. The global convergence rate of proposed method is no better than accelerated gradient descent.
2. There is no superlinear local convergence rates.
3. Assumption 2 is too strong.

**Reviewer Scores:**

The main concerns have not been addressed at rebuttal stage. Hence, I think the reviewers will not raise their scores.

---

### Decision · Program_Chairs · 2026-01-26

Reject